# *ACE*: All-round Creator and Editor Following Instructions via Diffusion Transformer

**Zhen Han**[*]**& Zeyinzi Jiang**[*] **& Yulin Pan**[*] **& Jingfeng Zhang**[*] **& Chaojie Mao**[*†]
Tongyi Lab, Alibaba Group

**Chenwei Xie & Yu Liu & Jingren Zhou**
Tongyi Lab, Alibaba Group

## Abstract

Diffusion models have emerged as a powerful generative technology and have been found to be applicable in various scenarios. Most existing foundational diffusion models are primarily designed for text-guided visual generation and do not support multi-modal conditions, which are essential for many visual editing tasks. This limitation prevents these foundational diffusion models from serving as a unified model in the field of visual generation, like GPT-4 in the natural language processing field. In this work, we propose **ACE**, an **A**ll-round **C**reator and **E**ditor, which achieves comparable performance compared to those expert models in a wide range of visual generation tasks. To achieve this goal, we first introduce a unified condition format termed Long-context Condition Unit (LCU), and propose a novel Transformer-based diffusion model that uses LCU as input, aiming for joint training across various generation and editing tasks. Furthermore, we propose an efficient data collection approach to address the issue of the absence of available training data. It involves acquiring pairwise images with synthesis-based or clustering-based pipelines and supplying these pairs with accurate textual instructions by leveraging a fine-tuned multi-modal large language model. To comprehensively evaluate the performance of our model, we establish a benchmark of manually annotated pairs data across a variety of visual generation tasks. The extensive experimental results demonstrate the superiority of our model in visual generation fields. Thanks to the all-in-one capabilities of our model, we can easily build a multi-modal chat system that responds to any interactive request for image creation using a single model to serve as the backend, avoiding the cumbersome pipeline typically employed in visual agents.

## 1 Introduction

In recent years, foundational generative models have made groundbreaking progress in natural language processing (NLP) (Anil et al., 2023; Anthropic, 2023a;b; Ouyang et al., 2022). Conversational language models like ChatGPT (Brown et al., 2020; OpenAI, 2023b) offer a unified framework for addressing various NLP tasks through a prompt-guided approach. By employing a unified input-output structure, these models can achieve dynamic multi-turn interactions with users. Furthermore, by harnessing the knowledge of historical dialogues (Anthropic, 2024; OpenAI, 2024), they possess the capacity to comprehend intricate queries with greater nuance and depth. However, such unified architecture has not been fully explored in visual generation field. Existing foundational models of visual generation typically create images or videos from pure text, which is not compatible with most visual generation tasks, such as controllable image generation (Zhang et al., 2023b; Jiang et al., 2024) or image editing (Brooks et al., 2023). Thereby, specific visual generation tasks still require tailored tuning based on these foundational models, which is inflexible and inefficient. For this reason, the visual generative model has not yet become a powerful and unified productivity tool in

---

[*]Equal Contribution. Order is determined by random dice rolling.
† Project leader and corresponding author.

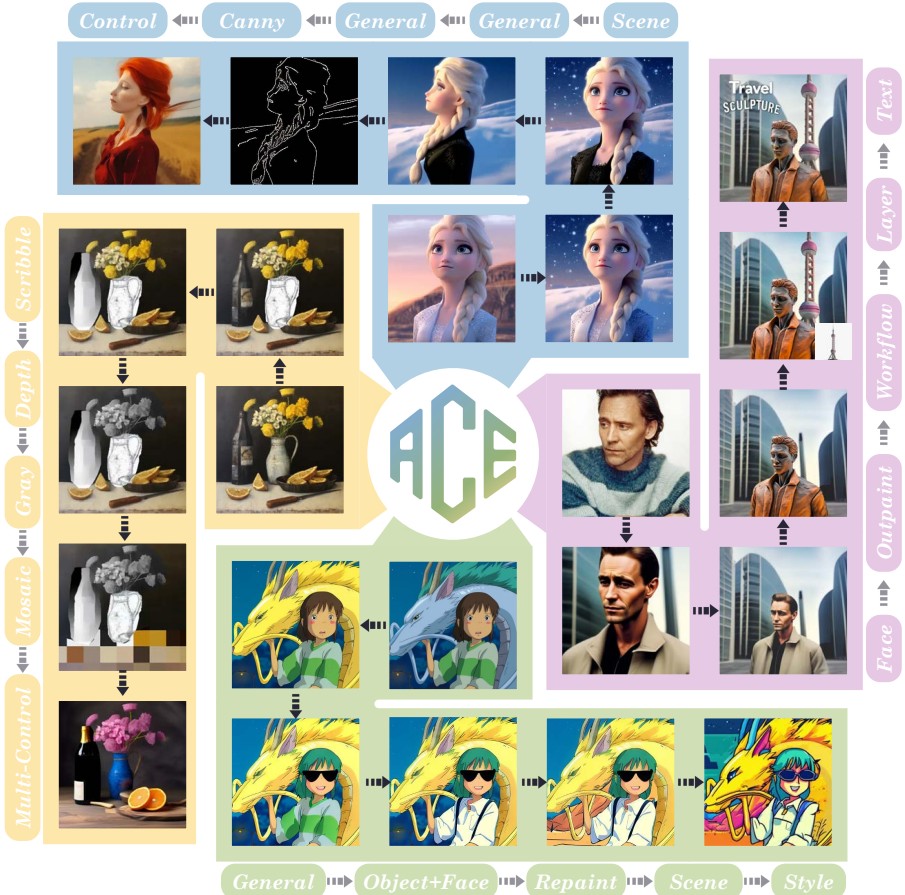

Figure 1: **Multi-turn image editing results of ACE**. ACE supports a wide range of image generation and editing tasks through natural language instructions, allowing complex and precise editing requests to be easily accomplished through multi-turn interactions.

various application scenarios like large language models (LLMs) (Abdin et al., 2024; Dubey et al., 2024; Bai et al., 2023; Yang et al., 2024).

One major challenge of building an all-in-one visual generation model lies in the diversity of multi-modal input formats and the variety of supported generation tasks. To address this, we design a unified framework using a Diffusion Transformer generation model that accommodates a wide range of inputs and tasks, empowering it to serve as an **A**ll-round **C**reator and **E**ditor, which we refer to as **ACE**. First, we analyze the condition inputs of most visual generation tasks, and define Condition Unit (CU), which establishes a unified input paradigm consisting of core elements such as image, mask, and textual instruction. Second, for those CUs containing multiple images, we introduce Image Indicator Embedding to ensure the order of the images mentioned in instruction matches image sequence within the CUs. Besides, we imply 3d position embedding instead of 2d spatial-level position embedding on the image sequence, allowing for better exploring the relationships among conditional images. Third, we concatenate the current CU with historical information from previous generation rounds to construct the Long-context Condition Unit (LCU). By leveraging this chain of generation information, we expect the model to better understand the user's request and create the desired image. As depicted in Fig. 1, ACE supports a range of generating and editing capabilities, allowing it to accomplish complex and precise generation tasks through multi-turn instructions.

To address the issue of the absence of available training data for various visual generation tasks, we establish a meticulous data collection and processing workflow to collect high-quality structured CU data at a scale of 0.7 billion. For visual conditions, we collect image pairs by synthesizing images from source images or by pairing images from large-scale databases. The former utilizes powerful open-source models to edit images to meet specific requirements, such as changing styles (Han et al., 2024) or adding objects (Pan et al., 2024), while the latter involves clustering and

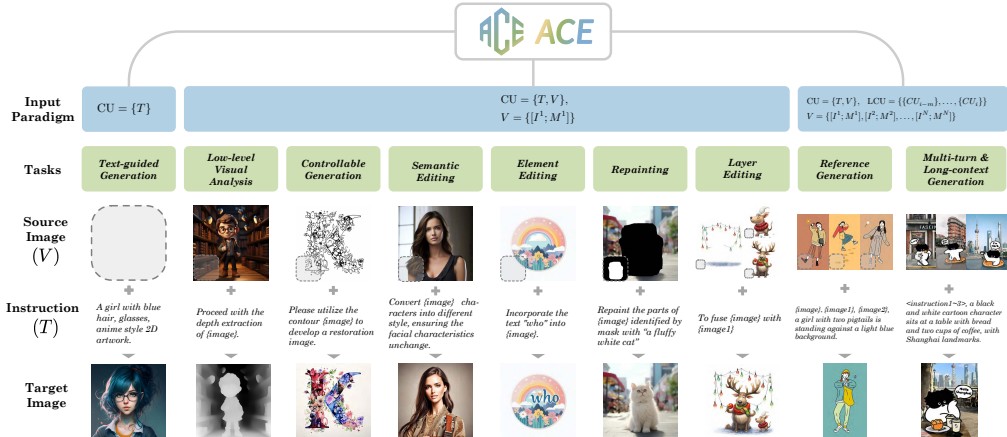

Figure 2: **The overview of all generation and editing task types supported by ACE**. These tasks are categorized into 8 basic types, multi-turn and long-context generation based on different input conditions (in green) and are formulated using the proposed input paradigm as 3 formats (in blue).

grouping images from extensive databases to provide sufficient real data, thereby minimizing the risk of overfitting to the synthesized data distribution. For textual instructions, we first manually construct instructions for diverse tasks by building templates or requesting LLMs, then optimize the instruction construction process by training an end-to-end instruction-labeling multi-modal large language model (MLLM) (Chen et al., 2024), thereby enriching the diversity of the text instructions.

Our ACE provides more comprehensive coverage of tasks on a single model compared to previous approaches. Therefore, to thoroughly evaluate the performance of our generation model, we construct an evaluation benchmark that encompasses the main tasks. This benchmark incorporates inputs sourced from both the real world and model-generated data, supporting global and local editing tasks. It is larger in scale and broader in scope compared to previous benchmarks (Sheynin et al., 2024; Zhang et al., 2023a). We conduct a user study to subjectively assess the quality of images generated by our method and the adherence to instructions, revealing that our approach generally aligns more closely with human perception across the majority of tasks. We summarize our main contributions as follows:

- We propose **ACE**, a unified foundational model framework that supports a wide range of visual generation tasks. To our knowledge, this is the most comprehensive diffusion generation model to date in terms of task coverage.

- By defining the CU for unifying multi-modal inputs across different tasks and incorporating long-context CU, we introduce historical contextual information into visual generation tasks, paving the way for ChatGPT-like dialog systems in visual generation.

- We design specific data construction pipelines for various tasks to enhance the quality and efficiency of data collection, and we ensure the richness of multi-modal data through MLLM fine-tuning for automated instruction labeling.

- We establish a more comprehensive evaluation benchmark compared to previous ones, covering the most known visual generation tasks. Evaluation results indicate that ACE demonstrates notable competitiveness in specialized models while also exhibiting strong generalization capabilities across a broader range of open tasks.

## 2 ALL-ROUND CREATOR AND EDITOR

ACE is an image creation and editing model based on the Diffusion Transformer that follows textual instructions. It establishes a unified framework that covers a wide range of tasks through the definition of standard input paradigm and strategy for aligning multi-modal information. With this exquisite design, the model is capable of handling various single tasks, multi-turn tasks, and long-context tasks with historical information.

## 2.1 PROBLEM DEFINITION

### 2.1.1 TASKS

When it comes to generation and editing, the input condition information varies significantly depending on the specific task types. This encompasses a diverse range of forms, including textual instructions, conditioning images in controllable generation, masks used in region editing, and images in guided generation, among others. We analyze and categorize these conditions from textual and visual modalities respectively: **(i) Textual modality**: we refer to all types of textual conditions as instructions and categorize them into **Generating-based Instructions** and **Editing-based Instructions**, depending on whether they describe the content of the generated image directly or the difference from the input visual cues; **(ii) Visual modality**: we categorize all generation tasks into 8 basic types, as shown in Fig. 2.

- **Text-guided Generation**. It only uses generating-based text prompt as a condition to create images, and none of the visual cues are adopted.
- **Low-level Visual Analysis**. It extracts low-level visual features from input images, such as edge maps or segmentation maps. One source image and editing-based instruction are required in the task to accomplish creation.
- **Controllable Generation**. It is the inverse task of Low-level Visual Analysis, which creates vivid images based on given conditions, *e.g.*, edge map, contour image, doodle image, scribble image, depth map, segmentation map, low-resolution image, *etc*.
- **Semantic Editing**. It aims to modify some semantic attributes of an input image by providing editing instructions, such as altering the style of an image or modifying the facial attributes of a character.
- **Element Editing**. It focuses on adding, deleting, or replacing a specific subject in the image while keeping other elements unchanged.
- **Repainting**. It erases and repaints partial image content of input image indicated by given mask and instruction.
- **Layer Editing**. It decomposes an input image into different layers, each of which contains a subject or background, or reversely fuses different layers.
- **Reference Generation**. It generates an image based on one or more reference images, analyzing the common elements among them and presenting these elements in the generated image.

By leveraging the generation tasks of these fundamental units, we can combine them to create **multi-turn scenarios**. Furthermore, utilizing the historical information from every round makes it possible to tackle **long-context visual generation** tasks.

### 2.1.2 INPUT PARADIGM

A significant obstacle to implementing different types of generation and editing task requests within one framework lies in the diverse input condition formats of tasks. To address this issue, we design a unified input paradigm defined as **C**onditional **U**nit (**CU**) that fits as many tasks as possible. The CUs composed of a textual instruction $T$ that describes the generation requirements, along with visual information $V$, where $V$ consists of a set of images $I$ that can be defined as $I = \emptyset$ (if there are no source image) or $I = \{I^1, I^2, \ldots, I^N\}$ (if there are source images) and corresponding masks $M = \{M^1, M^2, \ldots, M^N\}$. When there is no specific mask, $M$ is set to a blank image. The overall formulation of the CU is as follows:

$$\text{CU} = \{T, V\}, \quad V = \{[I^1; M^1], [I^2; M^2], \ldots, [I^N; M^N]\}, \tag{1}$$

where a channel-wise connection operation is performed between corresponding $I$ and $M$, $N$ represents the total number of visual information inputs for this task.

Furthermore, to better address the demands of complex long-context generation and editing, historical information can be optionally integrated into CU, which is formulated as:

$$\text{LCU}_i = \{\{T_{i-m}, T_{i-m+1}, \ldots, T_i\}, \{V_{i-m}, V_{i-m+1}, \ldots, V_i\}\} \tag{2}$$

where $m$ denotes the maximum number of rounds of historical knowledge introduced in the current request. $\text{LCU}_i$ is a **L**ong-context **C**ondition **U**nit used to generate desired content for the $i$-th request.

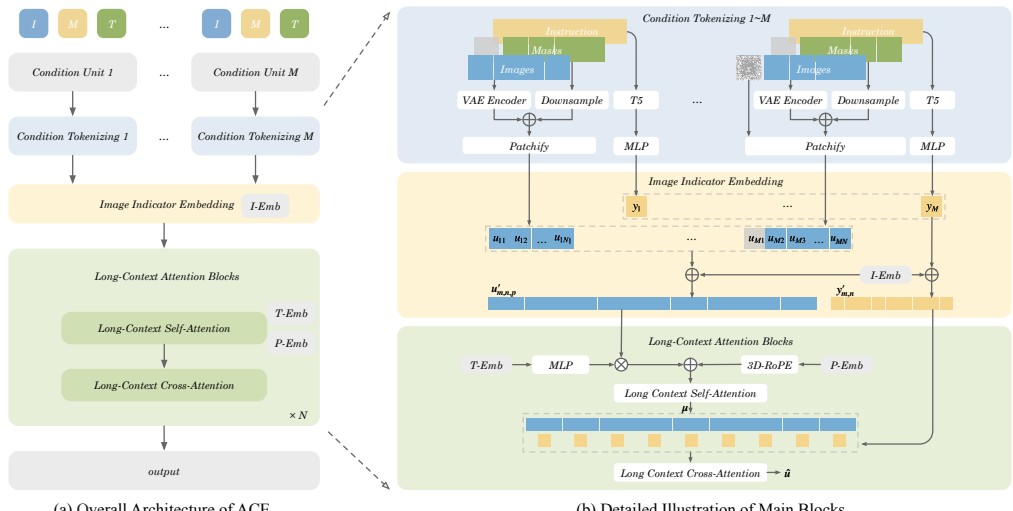

(a) Overall Architecture of ACE.

(b) Detailed Illustration of Main Blocks.

Figure 3: **The illustration of ACE framework.** Condition Tokenizing module tokenizes each input CU, concatenating them to obtain the visual token sequence and the text token sequence. The Image Indicator Embedding module employs pre-defined textual tokens to indicate the image order in textual instructions and distinguish various input images. The Long-context Attention Block ensures effective communication and integration of long-context sequences.

## 2.2 ARCHITECTURE

In this section, we introduce a unified visual generation framework that can perform all visual generation tasks within a single model, and incorporate long-context conditions to enhance comprehension. As illustrated in Fig. 3a, the overall framework is built based on a Diffusion Transformer model (Vaswani et al., 2017; Peebles & Xie, 2023), and integrated with three novel components to achieve unified generation: Condition Tokenizing, Image Indicator Embedding, and Long-context Attention Block. We will provide a detailed description of them below.

**Condition Tokenizing**. Considering an LCU that comprises $M$ CUs, the model involves three entry points for each CU: a language model (T5) (Raffel et al., 2020) to encode textual instructions, a Variational Autoencoder (VAE) (Kingma & Welling, 2014) to compress reference image to latent representation, and a down-sampling module to resize mask to the shape of corresponding latent image. The latent image and its mask (an all-one mask if no mask is provided) are concatenated along the channel dimension. These image-mask pairs are then patchified into 1-dimensional visual token sequences $u_{m,n,p}$, where $m$, $n$ are indexes for CUs and visual information Vs in each CU, while $p$ denotes the spatial index in patchified latent images. Similarly, textual instructions are encoded into 1-dimensional token sequences $y_m$. After processing within each CU, we separately concatenate all visual token sequences and all textual token sequences to form a long-context sequence.

**Image Indicator Embedding**. As illustrated as Fig. 3b, to indicate the image order in textual instructions and distinguish various input images, we encode some pre-defined textual tokens "{image}, {image1}, ..., {imageN}" into T5 embeddings as Image Indicator Embeddings ($I$-Emb). These indicator embeddings are added to the corresponding image embedding sequence and text embedding sequence, which is formulated as:

$$y'_{m,n} = y_m + I\text{-Emb}_{m,n}, \tag{3}$$

$$u'_{m,n,p} = u_{m,n,p} + I\text{-Emb}_{m,n}. \tag{4}$$

In this way, image indicator tokens in textual instructions and the corresponding images are implicitly associated.

**Long-context Attention Block**. Given the long-context visual sequence, we first modulate it with the time step embedding ($T$-Emb), then incorporate a 3D Rotational Positional Encodings (RoPE) (Su et al., 2023) to differentiate between different spatial- and frame-level image embeddings. During the Long Context Self-Attention, all image embeddings of each CU at each spatial location, are equivalently and comprehensively interact with each other by $\mu = Attn(u', u')$. Next,

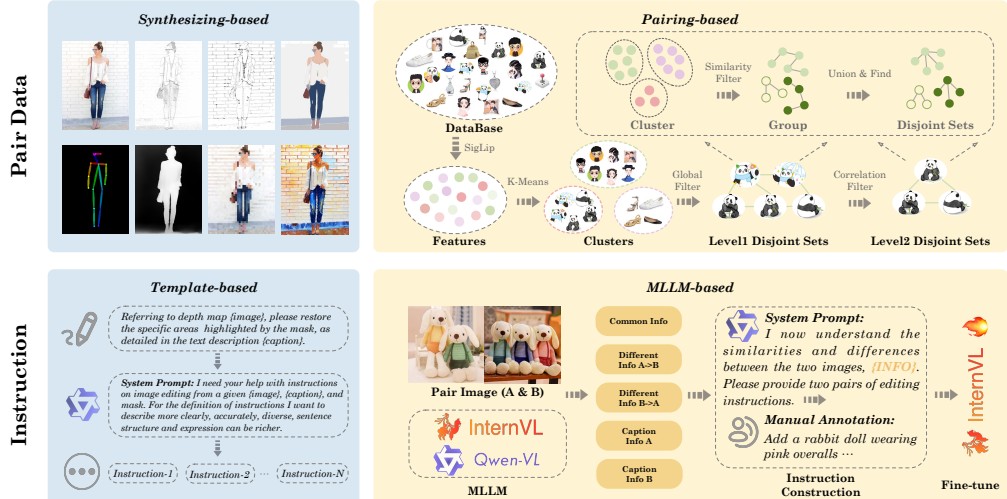

Figure 4: **The pipeline of dataset construction and instructions labeling.** In data construction, two methods are utilized: synthesizing using open-source expert models and mining from large-scale data. For instruction labeling, we combined templating with MLLM labeling, further training the Instruction Captioner to achieve large-scale instruction labeling.

unlike the cross-attention layer of the conventional Diffusion Transformer model, where each visual token attends to all of the textual tokens, we implement cross-attention operation with each condition unit. That means image tokens in $m$-th CU will only attend to the textual tokens from the same CU. This can be formulated as:

$$\hat{u}_{m,n} = Attn(\mu_{m,n}, y'_{m,n}). \tag{5}$$

This ensures that, within the cross-attention layer, the text embeddings and image embeddings align on a frame-by-frame basis.

## 3  DATASETS

### 3.1  PAIR DATA COLLECTION

A critical challenge of training foundational visual generation model lies in how to acquire pairwise images for various tasks. In this section, we introduce two ways to efficiently build high-quality datasets for most of the generation and editing tasks: **(i) Synthesizing from source image**: thanks to the rapid development in the field of visual generation, there have been many of powerful open-source models designed to solve one specific problem. Leveraging these powerful single-point technologies, we could synthesis plenty of image pairs for lots of generation and editing tasks, such as controllable generation, style editing, object editing, and so on. **(ii) Pairing from massive databases**: though the synthesis-based method is efficient and straightforward in acquiring pairwise data. However, It still possesses two drawbacks. First, some editing problems have not been fully explored, and there are no powerful open-source models available for these tasks. Second, using only synthetic data can easily cause over-fitting and reduce the quality of generated images. Therefore, it is essential to provide sufficient real data to address the aforementioned drawbacks. We propose a hierarchically aggregating pipeline for pairing content-related images from massive databases to build pairs of data for training, as illustrated in Fig. 4. We first extract semantic features using SigLIP (Zhai et al., 2023) from large-scale datasets (*e.g.*, LAION-5B (Schuhmann et al., 2022), OpenImages (OpenImage, 2023), and our private datasets). Then leveraging K-means clustering technology, coarse-grained clustering is implemented to divide all images into tens of thousands of clusters. Within each cluster, we implement a two-turn union-find algorithm to achieve fine-grained image aggregation. The first turn is based on the SigLIP feature and the second turn uses a similarity score tailored for specific tasks. For instance, we calculate the face similarity score for the facial editing task and the object consistency score for the general editing task. Finally, we collect all possible pairs from each disjoint set and implement cleaning strategies to filter high-quality pairs.

Benefiting from these two automatic pipelines, we construct a large-scale training dataset that consists of nearly 0.7 billion image pairs, covering 8 basic types of tasks, multi-turn and long-context generation. We depict its distribution in Fig. 6 and provide a detailed description of the specific data construction methods for each task, please refer to appendix B.

## 3.2 INSTRUCTIONS

In addition to collecting image pairs, it is essential to label clear natural language instructions that indicate how to transform one image into another. Compared to the caption generation commonly used in text-to-image task, instruction labeling is generally more challenging, as it requires analyzing not only the semantics of individual images, but also the discrepancies across multiple images. We employ both **Template-based** and **MLLM-based** methods to tackle this challenge. Template-based method constructs instruction templates for specific vision tasks by leveraging human knowledge priors. However, the instructions generated by this method lack diversity, which can lead to significant overfitting problems. MLLM-based method generates unique instructions for each given editing pair, leveraging off-the-shelf MLLMs. Nonetheless, current MLLMs exhibit limitations in producing precise instructions for editing tasks involving non-natural images, such as depth-controlled image generation and image segmentation. Thus, we combine these two methods and design an effective strategy to mitigate the aforementioned drawbacks. For tasks that contain non-natural images, we utilize a template-based method to generate instruction templates. These templates are then combined with the generated captions to produce the final instructions. To address the issue of insufficient diversity, we employ LLMs to reformulate instructions multiple times, and tune prompts to ensure that each rewritten version is distinct from all preceding instructions. For tasks that contain natural images, we employ an MLLM to predict the differences and commonalities between the images in the input pair. Then an LLM is used to generate instructions focusing on semantic distinctions according to the analysis of the differences and commonalities. Further, the collected instructions generated by these two methods undergo human annotation and correction. The revised instructions are used for fine-tuning an open-source MLLM, enabling it to predict instructions for any given image pair. Specifically, we collect a dataset of approximately 800,000 curated instructions and train an **Instruction Captioner** by fine-tuning the InternVL2-26B (Chen et al., 2024). Once trained, the Instruction Captioner is able to take any two images as input and generates the instruction for transforming the source image to the target image. It can also be further extended to the processing of cluster data, by entering a set of images, obtaining the similarity description among images within the cluster, and the differences between each pair within the cluster. The above process is illustrated in Fig. 4.

## 4 EXPERIMENTS

### 4.1 BENCHMARKS AND METRICS

**Existing Benchmarks**. We first evaluate on the commonly used benchmark MagicBrush (Zhang et al., 2023a). It contains an overall 1,053 edit turns and 535 edit sessions for single-turn and multi-turn image editing respectively. It compares the output images with groundtruth images and the provided target text descriptions. Following the setting proposed in the MagicBrush benchmark, we calculate the L1 distance, L2 distance, CLIP (Radford et al., 2021) similarity, DINO (Liu et al., 2023a) similarity between the generated image and groundtruth image, and CLIP similarity between the generated image and textual prompt. We also evaluate the Emu Edit benchmark (Sheynin et al., 2024), please see appendix F for details.

**ACE Benchmark**. To thoroughly evaluate the performance of various visual generation tasks, we build a benchmark dataset that covers all types of tasks the aforementioned. ACE benchmark consists of both real and generated images. The real images are primarily sourced from the MS-COCO (Lin et al., 2014) dataset and the generated images are created by Midjourney (Midjourney, 2023), using prompts obtained from JourneyDB (Sun et al., 2023a). For each task type, we manually craft instructions and masks to closely resemble actual user input patterns, reaching a total of 12,000 entries. The detailed statistics of ACE benchmark can be found in Fig. 24. We evaluate image quality and prompt following scores through a user study. The image quality score assesses the aesthetic quality of the generated images, while the prompt following score measures how well the images align with the provided textual instructions.

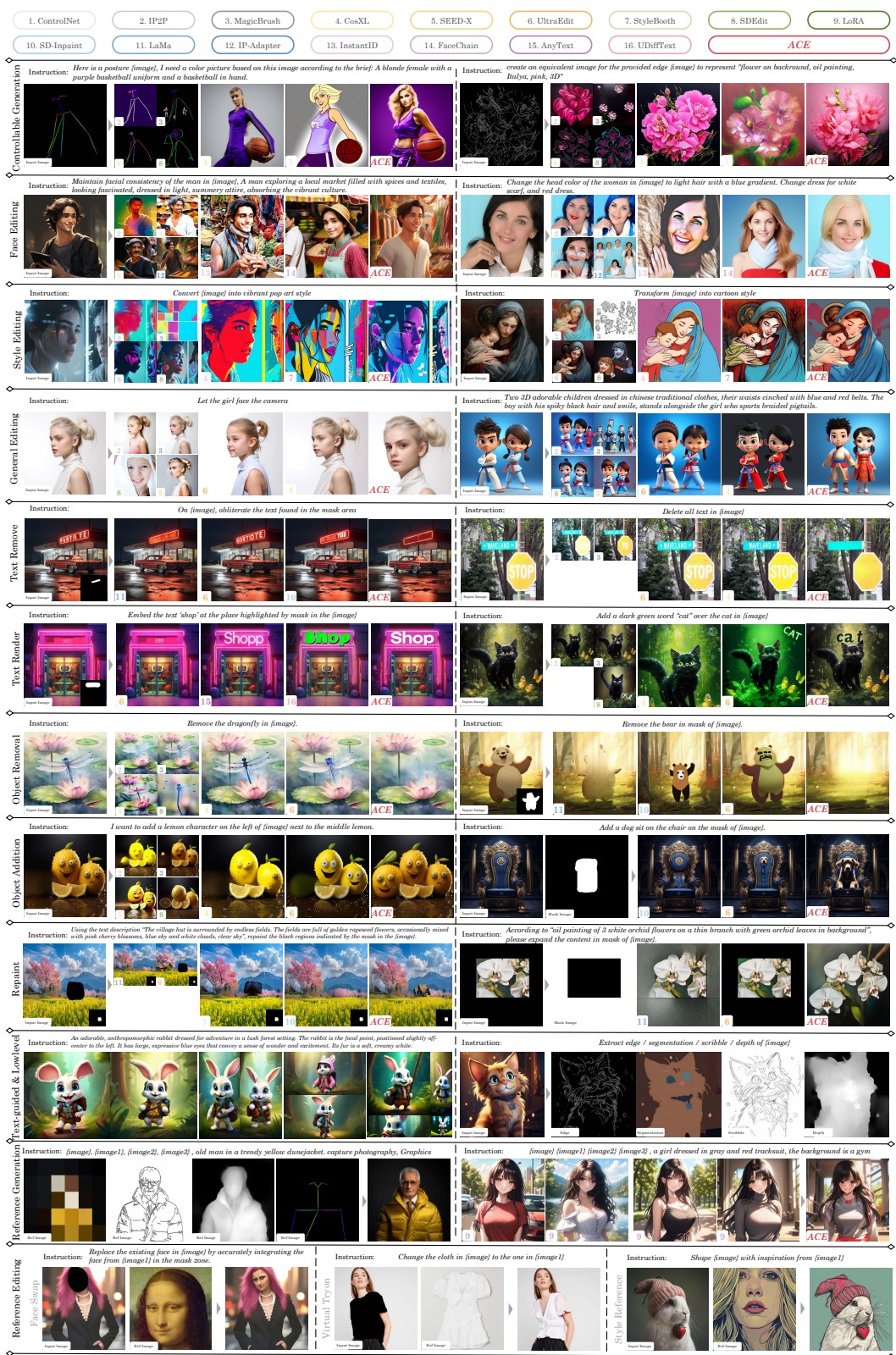

Figure 5: **Comparison and visualization of ACE performance with expert models** in different tasks. ACE demonstrates adaptability to multi-task and achieves superior performance.

Table 1: **Results on the MagicBrush benchmark**. LC denotes long-context generation with history.

| Settings | Methods | L1↓ | L2↓ | CLIP-I↑ | DINO↑ | CLIP-T↑ |
|---|---|---|---|---|---|---|
| | *Global Description-guided* | | | | | |
| Single-turn | SD-SDEdit (Meng et al., 2021) | 0.1014 | 0.0278 | 0.8526 | 0.7726 | 0.2777 |
| | Null Text Inversion (Mokady et al., 2022) | 0.0749 | 0.0197 | 0.8827 | 0.8206 | 0.2737 |
| | GLIDE (Nichol et al., 2022) | 3.4973 | 115.8347 | **0.9487** | **0.9206** | 0.2249 |
| | Blended Diffusion (Avrahami et al., 2022) | 3.5631 | 119.2813 | 0.9291 | 0.8644 | 0.2622 |
| | **ACE** (Ours) | **0.0505** | **0.0160** | 0.9436 | 0.9184 | **0.2833** |
| | *Instruction-guided* | | | | | |
| | HIVE (Zhang et al., 2024) | 0.1092 | 0.0380 | 0.8519 | 0.7500 | - |
| | InstructPix2Pix (Brooks et al., 2023) | 0.1122 | 0.0371 | 0.8524 | 0.7428 | 0.2764 |
| | MagicBrush (Zhang et al., 2023a) | 0.0625 | 0.0203 | 0.9332 | 0.8987 | 0.2781 |
| | UltraEdit (Zhao et al., 2024) | 0.0575 | 0.0172 | 0.9307 | 0.8982 | - |
| | **ACE** (Ours) | **0.0507** | **0.0165** | **0.9453** | **0.9215** | **0.2841** |
| | *Global Description-guided* | | | | | |
| Multi-turn | SD-SDEdit (Meng et al., 2021) | 0.1616 | 0.0602 | 0.7933 | 0.6212 | 0.2694 |
| | Null Text Inversion (Mokady et al., 2022) | 0.1057 | 0.0335 | 0.8468 | 0.7529 | 0.2710 |
| | GLIDE (Nichol et al., 2022) | 11.7487 | 1079.5997 | 0.9094 | 0.8494 | 0.2252 |
| | Blended Diffusion (Avrahami et al., 2022) | 14.5439 | 1510.2271 | 0.8782 | 0.7690 | 0.2619 |
| | **ACE** (Ours) | 0.0778 | 0.0290 | 0.9124 | 0.8611 | **0.2843** |
| | **ACE** (Ours w/ LC) | **0.0768** | **0.0285** | **0.9136** | **0.8635** | 0.2819 |
| | *Instruction-guided* | | | | | |
| | HIVE (Zhang et al., 2024) | 0.1521 | 0.0557 | 0.8004 | 0.6463 | 0.2673 |
| | InstructPix2Pix (Brooks et al., 2023) | 0.1584 | 0.0598 | 0.7924 | 0.6177 | 0.2726 |
| | MagicBrush (Zhang et al., 2023a) | 0.0964 | 0.0353 | 0.8924 | 0.8273 | 0.2754 |
| | UltraEdit (Zhao et al., 2024) | **0.0745** | **0.0236** | 0.9045 | 0.8505 | - |
| | **ACE** (Ours) | 0.0773 | 0.0293 | 0.9128 | 0.8661 | **0.2855** |
| | **ACE** (Ours w/ LC) | 0.0761 | 0.0284 | **0.9140** | **0.8668** | 0.2809 |

## 4.2 QUALITATIVE EVALUATION

In our qualitative evaluation, we present a comparison of our method with SOTA approaches across various tasks, including ControlNet (Zhang et al., 2023b), InstructPix2Pix (Brooks et al., 2023), MagicBrush (Zhang et al., 2023a), CosXL (StabilityAI, 2024), SEED-X Edit (Ge et al., 2024a), UltraEdit (Zhao et al., 2024), StyleBooth (Han et al., 2024), SDEdit (Meng et al., 2021), LoRA (Hu et al., 2022), SD-Inpaint (AI, 2022b), LaMa (Suvorov et al., 2022), IP-Adapter (Ye et al., 2023), InstantID (Wang et al., 2024b), FaceChain (Liu et al., 2023b), AnyText (Tuo et al., 2023), UDiff-Text (Zhao & Lian, 2024). In Fig. 5, we present qualitative comparisons between our single ACE model and 16 other methods across 12 subtasks. Overall, our method not only addresses a diverse range of tasks but also performs superior compared to task-specific methods. Additionally, we also show some extra tasks that the comparison methods do not perform well in the last three lines. Please see appendix H, for more examples of qualitative evaluation.

## 4.3 QUANTITATIVE EVALUATION

**Evaluation on Existing Benchmarks**. We first compare our method with baselines on the MagicBrush benchmark. Results are present on Tab. 1. For single-turn image editing, ACE significantly outperforms other methods under an instruction-guided setting while demonstrating comparable performance under a description-guided setting. For each setting of multi-turn image editing, we first employ the same inference way as MagicBrush, performing independent and continuous edits on a single image. The results show that our approach has significant advantages. Furthermore, we construct a long sequence using the historical information from each editing round, achieving a certain improvement in performance compared to not using it. This also demonstrates the effectiveness of LCU and architecture design.

**Evaluation on ACE Benchmark**. We conduct a comprehensive human evaluation using our benchmark to assess the performance of generated images, employing image scoring as the evaluation metric. Specifically, we score each image considering two aspects: prompt following and image quality. The prompt following metric measures the image compliance with text instructions or text descriptions, and is categorized into five levels. The image quality metric encompasses various aspects such as generated color, details, layout, and visual appeal, and is scored on a scale from 1 to 5. Considering the broad capabilities of our method, we compare it with several common approaches

Table 2: **User study results on ACE benchmark**. For each method in every supported task, we evaluate both prompt following and image quality, reporting the two scores in a single cell, separated by a "/". "-" means this task does not exist or is not supported by the current method.

| | Txt2img | Canny | Depth | Scribble | Pose | Face | Style | General | Add Text | Rm Text | Add Obj | Rm Obj | Inpaint | Outpaint |
|---|---|---|---|---|---|---|---|---|---|---|---|---|---|---|
| | | **Controllable** | | | | **Semantic** | | | **Element** | | | | **Repainting** | |
| *Global Editing* | | | | | | | | | | | | | | |
| SD1.5 (AI, 2022a) | 3.3/2.2 | - | - | - | - | - | - | - | - | - | - | - | - | - |
| SDXL (StabilityAI, 2022) | **4.1/2.8** | - | - | - | - | - | - | - | - | - | - | - | - | - |
| CtrlNet (Zhang et al., 2023b) | - | 2.5/2.0 | 3.8/2.4 | 1.9/2.0 | 2.9/1.9 | - | - | - | - | - | - | - | - | - |
| StyleBooth (Han et al., 2024) | - | - | - | - | - | - | **3.3**/2.6 | - | - | - | - | - | - | - |
| IP-Adapter (Ye et al., 2023) | - | - | - | - | - | 2.0/2.2 | - | 1.7/2.5 | - | - | - | - | - | - |
| InstantID (Wang et al., 2024b) | - | - | - | - | - | 2.5/2.7 | - | - | - | - | - | - | - | - |
| FaceChain (Liu et al., 2023b) | - | - | - | - | - | 2.0/3.0 | - | - | - | - | - | - | - | - |
| SDEdit (Meng et al., 2021) | - | 1.4/1.9 | 1.3/1.8 | 1.1/1.6 | 1.2/1.4 | 1.3/2.1 | 1.1/1.7 | 1.5/2.1 | 1.1/2.2 | 1.1/1.7 | 1.5/2.1 | 1.1/2.0 | - | - |
| IP2P (Brooks et al., 2023) | - | 1.9/2.0 | 1.7/2.0 | 1.5/2.3 | 1.4/1.4 | 2.3/2.4 | 2.4/2.5 | 2.2/2.4 | 1.1/2.6 | 1.3/2.6 | 2.0/2.4 | 1.5/2.4 | - | - |
| MB (Zhang et al., 2023a) | - | 1.3/1.8 | 1.3/1.7 | 1.3/1.9 | 1.1/1.3 | 2.4/2.3 | 1.4/2.0 | 2.2/2.3 | 1.5/2.4 | 2.2/2.5 | **3.1**/2.2 | 2.1/2.4 | - | - |
| SEED-X (Ge et al., 2024b) | - | 1.6/2.1 | 1.7/2.0 | 1.7/2.2 | 1.5/1.5 | 2.0/2.7 | 2.2/2.5 | 2.1/2.7 | 1.3/2.6 | 2.1/2.6 | 1.9/**2.6** | 2.5/2.4 | - | - |
| CosXL (StabilityAI, 2024) | - | 4.1/**2.9** | 4.1/**2.8** | 2.6/**2.9** | 3.7/2.1 | **2.9/3.1** | 3.2/**3.0** | 3.2/2.9 | 1.4/**2.7** | 1.0/**2.9** | 2.8/2.5 | 1.1/**3.1** | - | - |
| UltraEdit (Zhao et al., 2024) | - | 1.7/2.2 | 1.2/1.8 | 1.3/2.3 | 1.1/1.3 | 2.3/2.5 | 2.1/2.4 | 2.6/2.5 | 1.7/2.6 | 1.1/2.7 | 2.7/2.3 | 1.5/2.6 | - | - |
| **ACE (Ours)** | 3.7/2.5 | **4.6**/2.7 | **4.5/2.8** | **4.8/2.9** | **4.1**/2.3 | 2.8/2.8 | 2.4/2.6 | 2.1/2.5 | **2.8/2.7** | **4.4/2.9** | 2.6/2.4 | **3.9**/2.5 | - | - |
| *Local Editing* | | | | | | | | | | | | | | |
| LaMa (Suvorov et al., 2022) | - | - | - | - | - | - | - | - | - | 3.6/2.8 | - | **4.5/2.8** | 1.6/2.3 | 3.0/2.4 |
| SDInpaint (AI, 2022b) | - | - | - | - | - | - | - | - | - | 2.6/2.6 | 1.6/**2.7** | 2.2/2.5 | 3.6/2.6 | - |
| CtrlNet (Zhang et al., 2023b) | - | - | - | - | - | - | - | - | - | 2.9/2.7 | 1.9/2.5 | 2.6/2.2 | 3.0/2.1 | 3.2/2.1 |
| AnyText (Tuo et al., 2023) | - | - | - | - | - | - | - | - | 3.5/2.7 | - | - | - | - | - |
| UDiffText (Zhao & Lian, 2024) | - | - | - | - | - | - | - | - | 3.6/2.7 | - | - | - | - | - |
| UltraEdit (Zhao et al., 2024) | - | 1.4/1.9 | 1.2/1.8 | 1.2/2.0 | - | - | - | - | 1.1/2.8 | 1.2/2.9 | 2.9/2.5 | 1.4/2.5 | 1.1/1.7 | 1.1/2.1 |
| **ACE (Ours)** | - | 4.8/2.6 | 4.3/2.5 | 4.8/2.6 | - | - | - | - | 4.5/2.9 | 4.5/2.9 | 3.7/2.5 | 4.3/2.5 | 4.4/2.7 | 4.6/2.8 |

and some experts designed for specific tasks. We engaged 5 professional designers as evaluators to carry out these assessments. For each task, the data is evenly distributed among the evaluators in an anonymous manner, and scores are aggregated for analysis.

As shown in Tab. 2, we compare our approach across multiple global editing tasks and local editing tasks. The prompt following score and image quality score are presented together, separated by a "/" pattern. The bold numbers represent the best, and the underlined numbers indicate the second best. Our method achieves the highest prompt following scores in 7 of 12 global editing tasks and 8 of 10 local editing tasks, which demonstrates that ACE fully understands the intention of the instruction and is able to correctly generate an image that meets the instruction. Furthermore, ACE achieves the best image quality scores in 5 of 10 global editing tasks and 7 of 10 local editing tasks. These results indicate that ACE excels at generating high aesthetic images across various image editing tasks. Nonetheless, our method performs unsatisfactorily in certain tasks, such as general editing and style editing. One possible reason is that images generated by methods using larger models, such as those producing 1024-resolution images based on the SDXL model, are more preferred by evaluators compared to those produced by our model, which has a size of 0.6B parameters and an output resolution of around 512.

## 5 CONCLUSION

We propose ACE, a versatile foundational generative model that excels at creating images, and following instructions across a wide range of generative tasks. Users can specify their generation intentions through customized text prompts and image inputs. Furthermore, we advance the exploration of capabilities within interactive dialogue scenarios, marking a significant step forward in the processing of long contextual historical information in the field of visual generation. Our work aims to provide a comprehensive generative model for the public and professional designers, serving as a productivity enhancement tool to foster innovation and creativity.

**Acknowledgments.** We sincerely appreciate the contributions of many colleagues for their insightful discussions, valuable suggestions, and constructive feedback, including: Haiming Zhao, Yuntao Hong, You Wu, Jixuan Chen, Yuwei Wang, and Sheng Yao for their data contributions, and Lianghua Huang, Kai Zhu, and Yutong Feng for their discussions, suggestions, and the sharing of resources.

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
