# Table of Contents for ACE

## A    RELATED WORK

Visual generation, which takes multi-modal conditions (*e.g.*, textual instruction and reference image) as input to generate creative image, has emerged as a popular research trend in recent years. As the basic task, text-guided image generation has undergone a significant development, marked by remarkable advancements in recent years. Many approaches (Nichol et al., 2022; Saharia et al., 2022; OpenAI, 2022; Rombach et al., 2022; StabilityAI, 2022; OpenAI, 2023a; Midjourney, 2023; Cloud, 2023; Zhang et al., 2021; Chen et al., 2023a; Esser et al., 2024; KOLORS, 2024; Li et al., 2024; FLUX, 2024) have been proposed and achieved impressive results in terms of both image quality and semantic fidelity. By incorporating low-level visual features as input, Huang et al. (2023) and Zhang et al. (2023b) pave the way for the initial forms of multi-modal controllable generation. Recently, some approaches (Mou et al., 2023; Zhao et al., 2023; Qin et al., 2023) have tried to use multiple visual features as conditions, facilitating the multi-modal controllable generation. By integrating fine-tuning technologies such as Ruiz et al. (2023); Hu et al. (2022), these approaches have further enabled the customization of diverse controllable generation applications. Another popular trend is image editing technology (Ye et al., 2023; Han et al., 2024; Wang et al., 2024b; Huang et al., 2024a; Wang et al., 2024a; Liu et al., 2023b; Tuo et al., 2023; Chen et al., 2023b; Pan et al., 2024; Wang et al., 2023; Xie et al., 2023; Sun et al., 2023b; Huang et al., 2024b; Bodur et al., 2024; Shi et al., 2024; Li et al., 2023; Meng et al., 2021), which focus on editing input images according to text prompts and preserving some identity such as person, scene, subject, or style. While the above models excel at generating image in one specific task or scenario, they have difficulty in extending to unseen tasks. To address the aforementioned challenges, some methods have been introduced to edit input images by following natural language instructions (Brooks et al., 2023; StabilityAI, 2024; Geng et al., 2024; Sheynin et al., 2024; Zhao et al., 2024; Ge et al., 2024b) which is more flexible to implement various tasks within a single model. However, a key bottleneck for these methods lies in the construction of high-quality instruction-paired datasets with annotated edits, which cause limited generalizability and suboptimal performance. In this paper, we focus on establishing a unified definition for multi-modal generation problems. Based on this definition, we aim to construct higher-quality, annotated data and instruction sets further to develop a unified foundational model for multimodal generation.

## B    DATASETS DETAIL

We use an internal dataset of 0.7 billion data pairs to train a foundational model for generation and editing. The supported tasks include **8** basic types consisting of **37** subtasks, as well as a multi-turn and long-context generation task. These tasks use textual instructions along with zero or more reference images for generating or editing image. The data distribution is depicted in Fig. 6a, and the absolute data scale is illustrated in Fig. 6b. In this section, we provide a detailed introduction to the data construction methods for various tasks.

### B.1    TEXT-GUIDED GENERATION

We collect approximately 117 million images and use MLLM model to supplement captions for images, creating pair data for text-to-image tasks. Additionally, this portion of the data serves as an intermediary bridge in various generation and editing tasks, allowing the combination of different task instructions to obtain pairs from original images to target images.

### B.2    LOW-LEVEL VISUAL ANALYSIS

Low-level Visual Analysis tasks involve analyzing and extracting various low-level visual features from a given image, like an edge map or segmentation map. These low-level visual features are typically employed as control signals in the controllable generation. We select 10 commonly used low-level features in the controllable generation, including segmentation map, depth map, human pose, mosaic image, blurry image, gray image, edge map, doodle image, contour image, and scribble image. The visual features extracted at global and local levels are illustrated in Fig. 7 and Fig. 8, respectively.

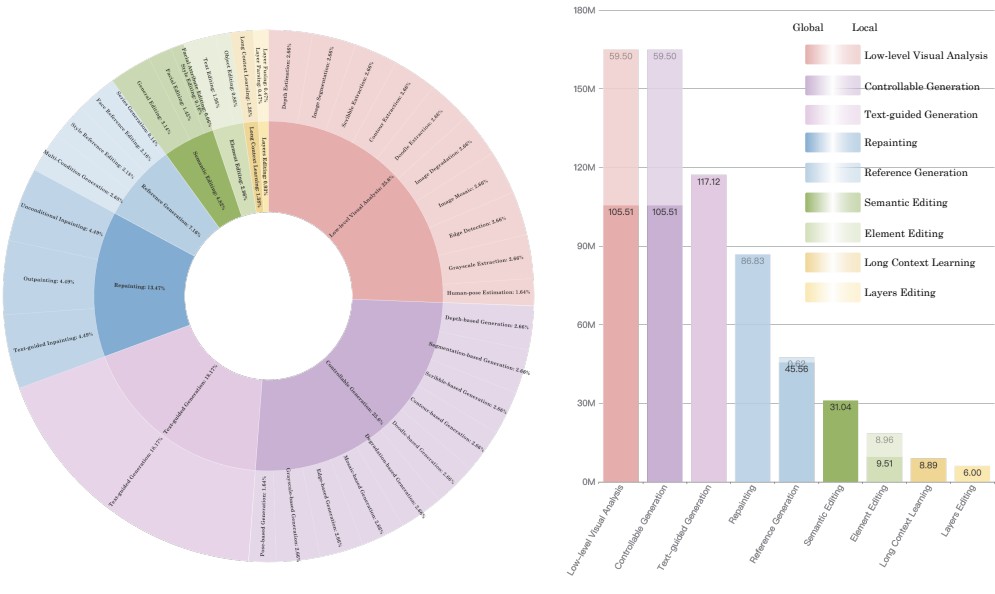

a. The distribution of all tasks in the dataset

b. The data scale of basic tasks in the dataset

Figure 6: **Statistics on the data scale for various tasks.** We collect 0.7 billion data pairs, which cover **8** basic types including **37** subtasks, multi-turn and long-context generation datasets.

- **Image Segmentation** involves extracting image spatial region information for different targets within an image. This is achieved by selecting and modifying specific areas for operations and editing in downstream tasks. We employ the Efficient SAM (Xiong et al., 2023) tool for marking different target areas within an image.

- **Depth Estimation** indicates the relative distance information of different targets within an image. We use the Midas (Ranftl et al., 2022) algorithm to extract depth information.

- **Human-pose Estimation** is employed for modeling the human body to obtain structured information about body posture. We make use of the RTMPose (Jiang et al., 2023) algorithm to extract information from images containing human figures, and posture information visualization is done using OpenPose's 17-point (Cao et al., 2021) modeling method.

- **Image Mosaic** pixelates specific areas or the entire image to protect sensitive information.

- **Image Degradation** is used to degrade the quality of an image to simulate the phenomenon of image distortion found in the real world. Following the practice of super-resolution algorithms (Wang et al., 2021), we add random noise to the input images.

- **Image Grayscale** is typically done to facilitate the editing of an image's original colors downstream. We do this conversion directly using OpenCV's Grayscale function.

- **Edge Detection** detects the edge information from the original image. We utilize the edge detection method named Canny (Canny, 1986) implemented by OpenCV.

- **Doodle Extraction** is usually used to simulate relatively rough hand-drawn sketches by extracting the outline of objects and ignoring their details. We use the PIDNet (Xu et al., 2023) and SketchNet (Zhang et al., 2016a) to extract this information.

- **Contour Extraction** is about delineating the outline of targets within an image, which simulates the drawing process of the image and is often used for secondary processing of images. We use the contour module from the informative drawing (Chan et al., 2022) for this information extraction.

- **Scribble Extraction** involves retrieving the original line art information to capture the sketch-like form of the image. We utilize the anime-style module from informative drawings (Chan et al., 2022) to extract the relevant information.

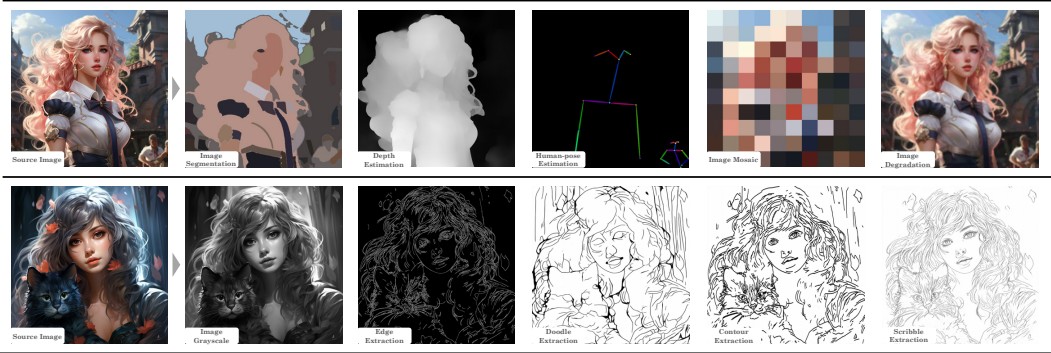

Figure 7: The visualization of low-level visual analysis preprocessing.

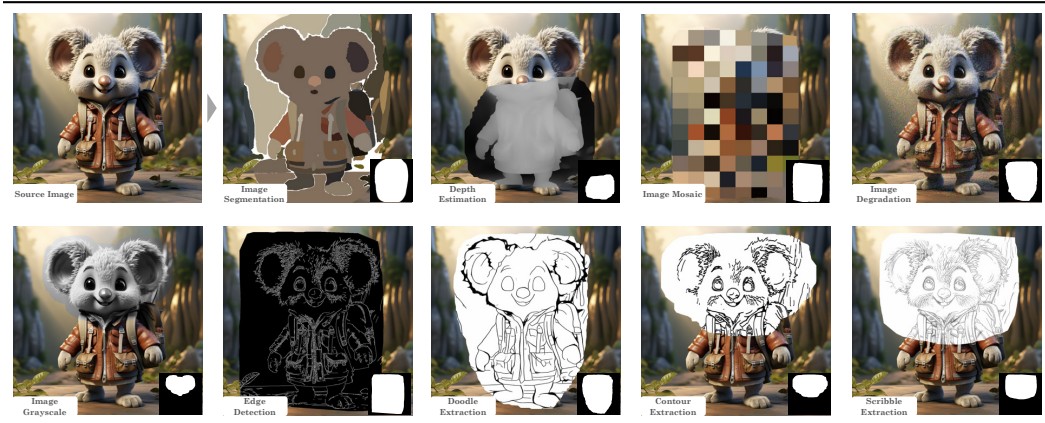

Figure 8: The visualization of regional low-level visual analysis preprocessing.

### B.3 CONTROLLABLE GENERATION

In the realm of vision-based generative foundation models, the ability to generate corresponding content using any provided prompts is commonly present. To further control aspects such as spatial layout, structure, or color in the generated images, additional conditional information is often incorporated as inputs to the model. We integrate various controllable condition-to-image tasks within a unified framework to accommodate different visual conditions. The control conditions include the visual features mentioned in the low-level visual analysis section. For training data, we employ pairs constituted by the aforementioned control conditions in Fig. 7 and regional control conditions in Fig. 8 obtained through low-level visual analysis, using the conditional part as inputs to the model to achieve pixel-precise image generation. For text guidance, we construct the instructions based on image captions with our proposed Instruction Captioner.

### B.4 SEMANTIC EDITING

Semantic Editing aims to modify specific semantic attributes of an input image by providing detailed instructions. It involves facial editing, which aims to modify partial attributes of characters while preserving the overall identity, and style transforming, which aims to transform the image style to a specific artist theme guided by instruction while keeping content unchanged. Additionally, any other semantic editing requests that do not fall into these two categories are classified as general editing, *e.g.*, changing the background scene of an image, adjusting a subject's posture, and modifying the camera view. We discuss the specifics according to the particular tasks.

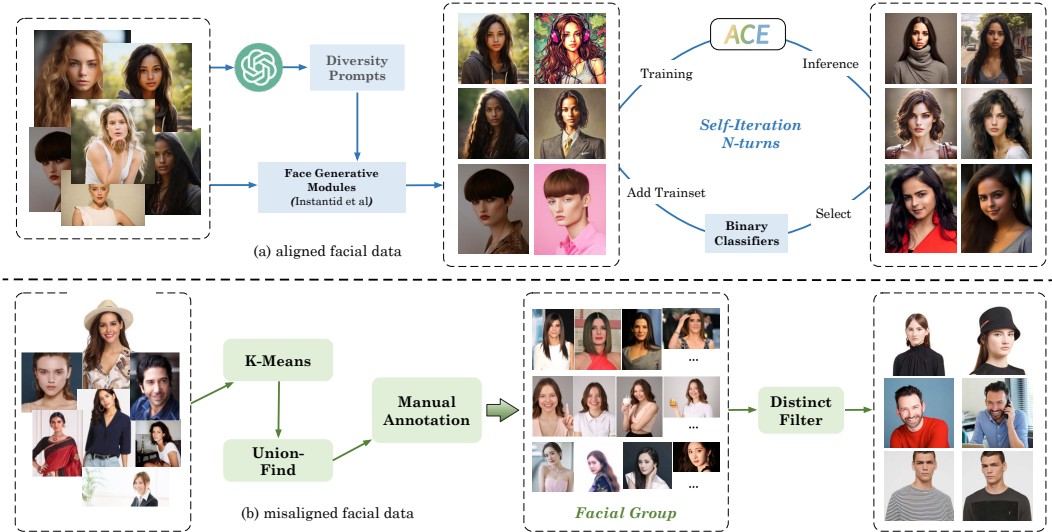

Figure 9: Illustration of facial editing data processing workflow.

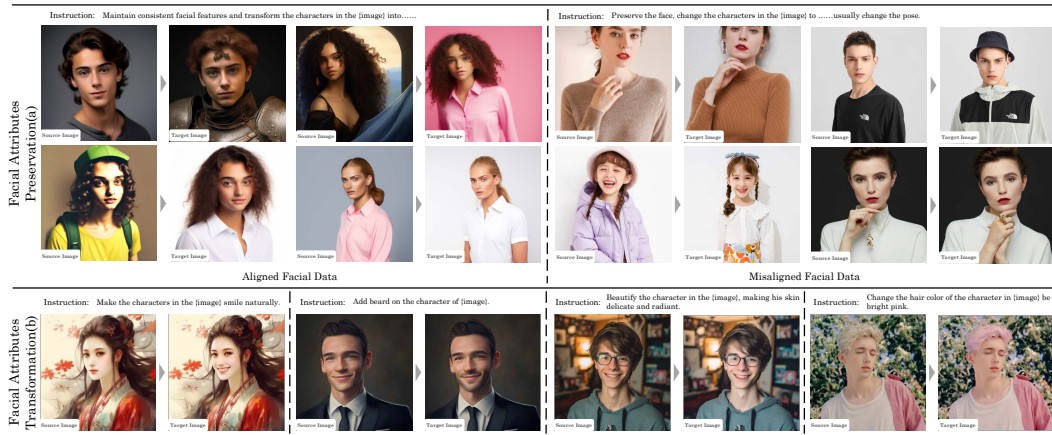

Figure 10: The dataset visualization of facial editing.

### B.4.1 FACIAL EDITING

Facial Editing encompasses both the transformation and preservation of facial attributes. Specifically, the facial attributes preservation task focuses on editing other elements of the image while maintaining the consistency of complex identity details in facial representations. The facial attributes transformation task is primarily concerned with altering specific attributes of the face without affecting other aspects of the image.

**Facial Attribute Preservation.** The facial attribute preservation dataset is divided into two main parts: aligned and misaligned facial data as shown in Fig. 10a. There are two novel processing workflows as shown in Fig. 9. **(i) Aligned facial data.** We generate pixel-aligned face data using generative models such as InstantID (Wang et al., 2024b) and combine it with GPT models to produce diverse prompts. Subsequently, we train multiple lightweight binary classification models to clean the generated data based on image quality, PPI score, aesthetic scores, and other metrics. Additionally, we extract facial features using ArcFace (Deng et al., 2019a) for similarity calculations, selecting high-matching data pairs with a similarity score exceeding 0.65. Once our model demonstrates the ability to maintain facial integrity, we initiate a self-iterative training process to generate higher quality data, as illustrated in Fig. 9a. **(ii) Misaligned facial data.** We first employ a face detection algorithm (Zhang et al., 2016b) to filter images containing only one face. Subsequently, we utilized facial features to perform K-means clustering, resulting in 10,000 clusters. Within each

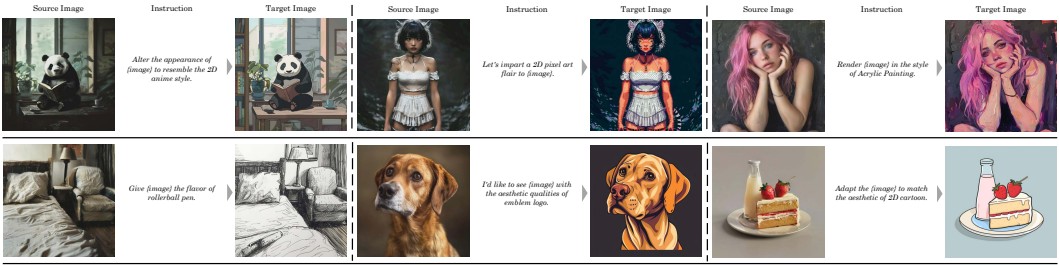

Figure 11: The dataset visualization of style editing.

cluster, we conducted a second clustering using the union-find algorithm. Faces with a similarity score greater than 0.8 and less than 0.9 were linked to avoid grouping perfectly identical images. Finally, manual annotation and deduplication were performed on the remaining clusters, yielding the final unaligned facial dataset as shown in Fig. 9b. Based on the general instruction construction process in Sec. 3.2, we design the instructions for facial editing to emphasize that the individuals in the image pairs being annotated are the same person. The instructions must reflect this and focus on the differences in personal details between the two images.

**Facial Attribute Transformation.** We add four fine-grained facial attribute transformation tasks: smiling, beard, makeup, and hair dyeing. We obtained the relevant data in bulk by calling the Aliyun API and trained binary classifiers for each category to filter out data with indistinct changes. As a result, we acquired a total of 1.4 million high-quality pairs of data as shown in Fig. 10b. Equally, we strive to guide the generated captions to closely reflect the facial attributes, thereby enhancing the model's understanding of the similarities and differences in tasks related to facial attributes.

### B.4.2 STYLE EDITING

Following the similar image pair construction strategy from StyleBooth (Han et al., 2024), we prepare a larger training data that encompasses over 80 styles and 63000 image pairs. Besides, additional real-world and synthesized style images are collected as style editing target images, and their corresponding "original" images are generated by transforming these collected images to different graphic styles such as cinematic, photography, *etc*. In this way, we obtain around 70000 input and output image pairs of about 400 high-quality styles. We show samples of the final style editing data in Fig. 11.

We conduct different filter strategies to leverage the data quality: (i) Like StyleBooth, we use CLIP score as the metric to filter out the image pairs which have too minor or too great differences. (ii) To further filter out the faultful synthesized target images that are not particularly aligned with the provided prompt keywords in terms of style, we use CSD (Somepalli et al., 2024) representations and implement style clustering within every style subgroup. Setting a threshold of 0.65, cosine similarities are calculated for union-find clustering. The largest cluster contains images in a similar visual style while other clusters are filtered out.

### B.4.3 GENERAL EDITING

The objective of general editing is to curate an image that seamlessly harmonizes with both textual and visual prompts for a variety of purposes. It involves two tasks, *i.e.*, caption-guided image generation and instruction-guided image adaption. The former task receives one reference image and one caption as prompts to generate the image, and the latter task intends to adapt the source image by following the given instructions. The training data for these two tasks can be unified into the same format, which consists of a content-related image pair $(I_{source}, I_{target})$, and a text prompt indicates how to generate target image. An essential goal of building such a training dataset is to acquire content-related image pairs, one of which serves as the source image and another serves as target image. The overall dataset construction pipeline is depicted in Fig. 12. It includes two branches, *i.e.*, **clustering-based** method, and **synthesis-based** method.

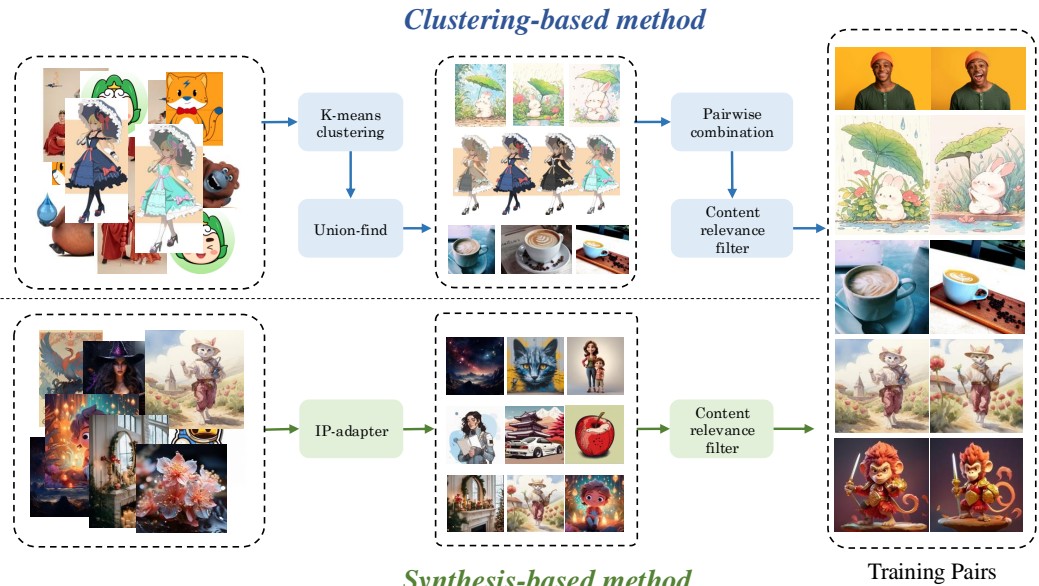

Figure 12: The dataset construction pipeline for general editing task.

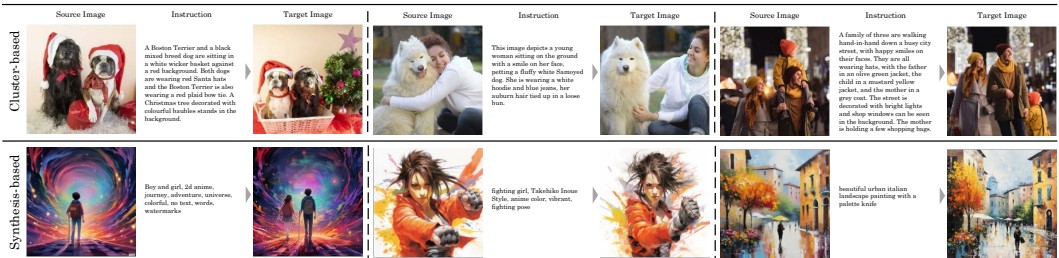

Figure 13: **General editing sample pairs generated by our dataset construction pipeline.** Image pairs in the first row are generated by cluster-based method, and those in the second row are generated from synthesis-based method.

**Cluster-Based Method.** We employ embedding-based clustering on the database to group content-similar images. Union-find technology is employed inside each cluster to achieve more fine-grained image pair aggregation. We then collect all possible pairs from each disjoint set. Additionally, a binary classifier evaluates the content relevance of pairs, and those with low relevance are discarded.

**Synthesis-Based Method.** We use IP-Adapter technology to synthesize images according to the reference images and text prompts, thus the content-related image pairs can be obtained. To ensure visual content is similar but not the same, we set the image control strength $\lambda$ to 0.6, and a binary classifier is utilized to filter out content-unrelated pairs. We depict some generated samples in Fig. 13.

For the text prompt of each image pair, we use the MLLM to generate both a caption that describes the visual content of the target image, and an instruction that indicates how to adapt the source image to the target image, as described in Sec. 3.2.

### B.5 ELEMENT EDITING

Element editing focuses on the selective manipulation of specific subjects within an image. This process allows for the addition, deletion, or replacement of a particular subject while ensuring that the other elements within the image remain unchanged. By doing so, the integrity of the overall com-

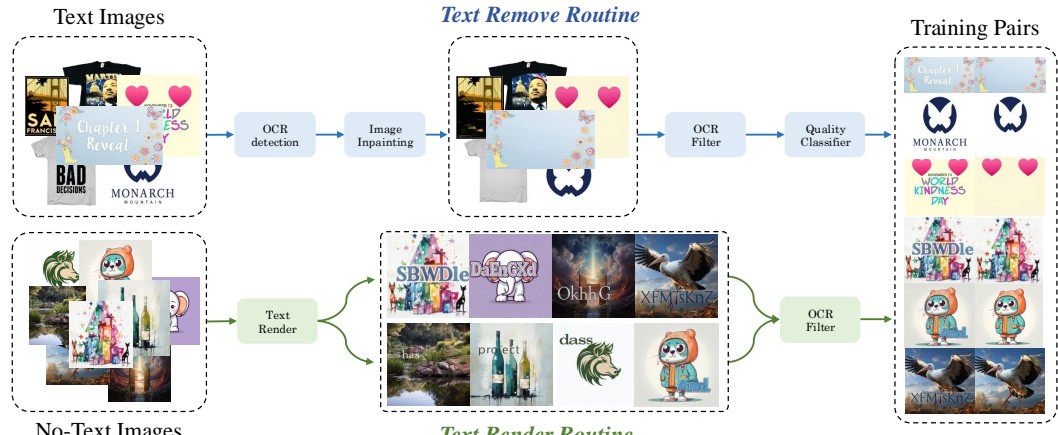

Figure 14: The pipeline of building training data of text editing.

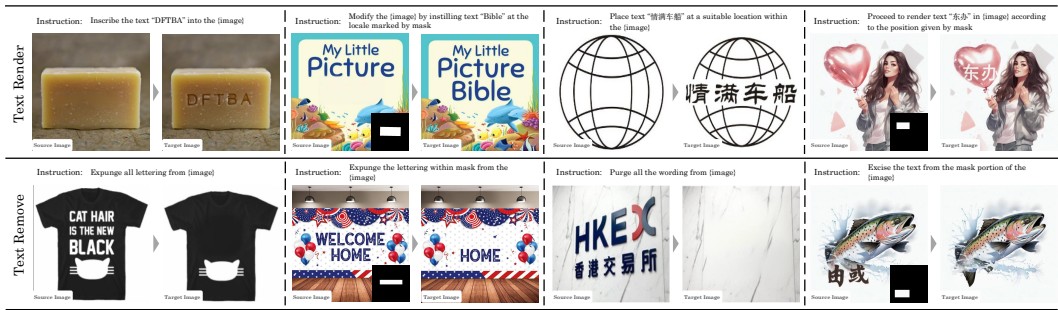

Figure 15: The dataset visualization of multi-lingual text editing.

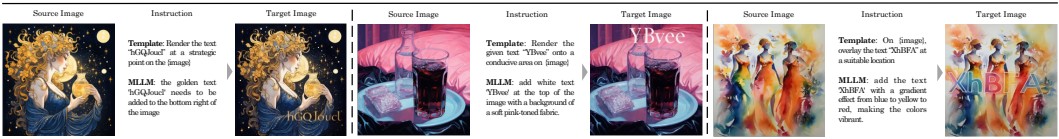

Figure 16: Template-based instructions and MLLM-based instructions on text editing.

position is preserved, allowing users to make precise edits and achieve desired alterations without disrupting the context of the original scene. We focus on two common elements: text and objects.

### B.5.1 TEXT EDITING

Text editing is an important task of element editing. Despite the progress gained in image generation, the capability of text rendering is still far from satisfying. Therefore, text editing is a necessary technology to revise the incorrect or deformed text rendered in image. Text editing involves text removing task, which is to erase text from image while preserving all other visual cues, and text rendering task, which is to render specific text at any location of an image. The goals of these two tasks are exactly the opposite, hence their training data can be shared to each other. For instance, for any image pair $\{I_a, I_b\}$, suppose the text removing represents the generation direction from $I_a$ to $I_b$, on the contrary, the generation direction from $I_b$ to $I_a$ stands for text rendering. Therefore, the objective of constructing the dataset thus becomes how to obtain a large number of image pairs, where one image contains the specified text and the other does not while keeping the non-text content unchanged.

We propose a two-branch data collection method to address this issue. The overall pipeline shown in Fig. 14 includes two paths: (i) **Text remove path**. For images containing text data, which typi-

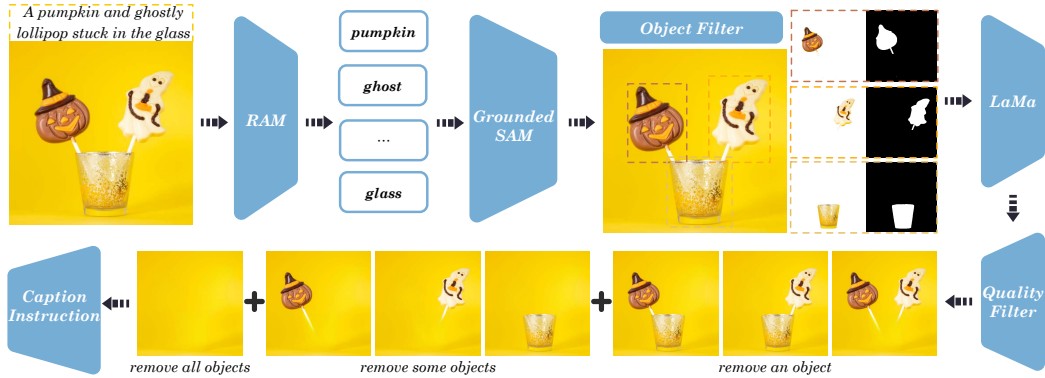

Figure 17: Illustration of data construction pipeline for object editing in element editing.

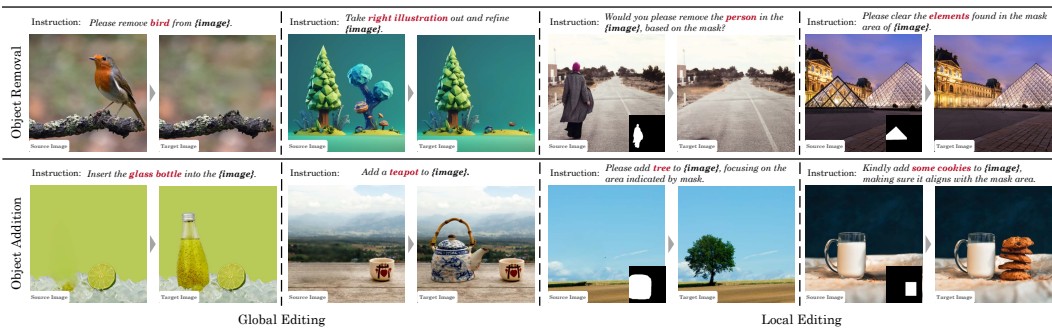

Figure 18: The dataset visualization of object editing in element editing.

cally from publicly available text datasets such as AnyWord3M (Tuo et al., 2023) and LaionGlyph-10M (Yang et al., 2023), we first mask out all text regions. Then, we redraw the masked areas leveraging image inpainting method. To ensure the regenerated image does not contain any textual information, we employ OCR detection leveraging the open-sourcing OCR model (*e.g.*, PP-OCR) (Du et al., 2020) and filter out all images that contain any texts. Finally, we adopt an image quality score predictor which is trained with small amounts of manually annotated data to score all text-removed images and pick high-quality samples according to the score. (ii) **Text render path**. For any image dataset, We first employ OCR detection to ensure input images contain no text. Then random characters are rendered in random locations of these images by utilizing existing text editing methods (*e.g.*, AnyText) (Tuo et al., 2023). We render text using Chinese or English characters to support multi-lingual text rendering capability. We depict some cases in Fig. 15. Finally, we implement OCR detection on the edited image to ensure all characters are rendered correctly. When training, image pairs collected from both two paths are merged to form the total dataset.

We adopt template-based and MLLM-based methods to construct instructions that describe how to render or remove text from the input image. For MLLM-based method, besides the content of characters, we add extra color and position controls by specifying the text color and render position in the textual instruction. Given a text image, we utilize a pre-trained MLLM to describe the color, content, and position information of text in this image, thus a text editing instruction can be easily inferred based on these descriptions. Some cases of template-based and MLLM-based instructions are illustrated in Fig. 16.

### B.5.2 OBJECT EDITING

Object-based image editing is one of the most commonly used techniques for creatively manipulating images. Its primary goal is to either remove or add objects in an image based on text instructions provided by the user, while ensuring a harmonious composition. To obtain training data for this task, we need to construct a pair of data to indicate the presence relationship of objects. Specifically, we

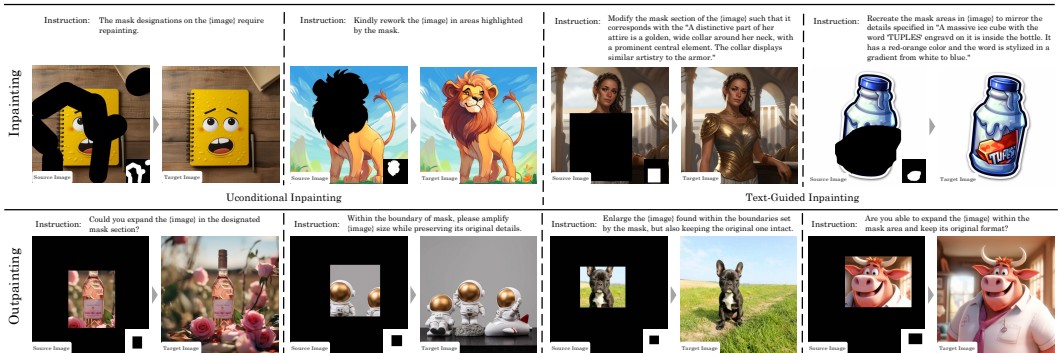

Figure 19: The dataset visualization of Repainting.

focus on images that either contain a specific object or do not, ensuring that all other parts of the images remain as unchanged as possible, except for the area where the object is located.

We can see the entire dataset process from Fig. 17. We first utilize RAM (Zhang et al., 2023c) for open-label tagging, obtaining semantic labels for different subjects in the image, and then applying Grounded-SAM (Liu et al., 2023a; Kirillov et al., 2023) to segment the input semantics. Next, we perform a preliminary screening of objects based on filtering criteria including the area of the masks and bounding boxes, as well as their effective ratios, removing any unreasonable subjects. We then use the LaMa (Suvorov et al., 2022) method, combining the original image with the subject mask area for inpainting. This operation effectively removes local objects without significantly affecting other areas. Finally, we employ a pre-trained binary classification model to determine whether the inpainted image meets expectations, filtering out artifacts introduced by the inpainting algorithm. In terms of instruction formulation, we employ a template format that incorporates the {object_name} tag, while also utilizing a common instruction based on image pairs.

Through the data construction pipeline, we can obtain the original image, the image with the object removed, the object mask, and the corresponding text instructions. This way, we can implement a forward pipeline for object removal and a reverse pipeline for object addition, while ensuring the integrity of the image and the accuracy of the text instructions, as in Fig. 18.

### B.6 REPAINTING

The repainting task can be defined as the process of reconstructing missing image information within specified masked regions. Depending on the location of the masked area and input conditions, this task can be categorized into three distinct types: unconditional inpainting, text-guided inpainting, and outpainting. Some examples of training data are shown in Fig. 19.

#### B.6.1 UNCONDITIONAL INPAINTING

Unconditional image inpainting typically utilizes methods such as low-level textual information and Fourier Convolutions, combined with contextual information from the known areas of the image, to reconstruct the missing portions. This process usually requires an input consisting of an image to be inpainted and a mask indicating the regions that need to be filled, leading to an output image where the missing areas are completed. The task demands that the original information is preserved and that there is a high-quality seamless integration between the original and the filled-in areas. By employing LaMa's (Suvorov et al., 2022) mask generation strategy, we randomly apply bbox or irregular-shaped masks to the images and vary the degree of this operation to enable the model to handle different types of missing regions as effectively as possible.

#### B.6.2 TEXT-GUIDED INPAINTING

Text-guided inpainting primarily aims to fill and restore missing parts of an image by utilizing text descriptions to guide the process. Unlike traditional unconditional inpainting, this method integrates textual information to guide the model, resulting in images that better meet the user's specific

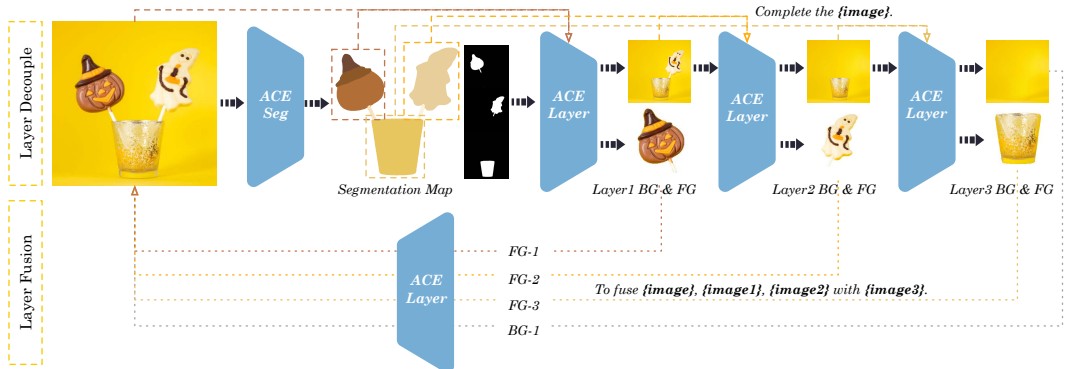

Figure 20: Illustration of inference pipeline layer decouple and layer fusion in layer editing.

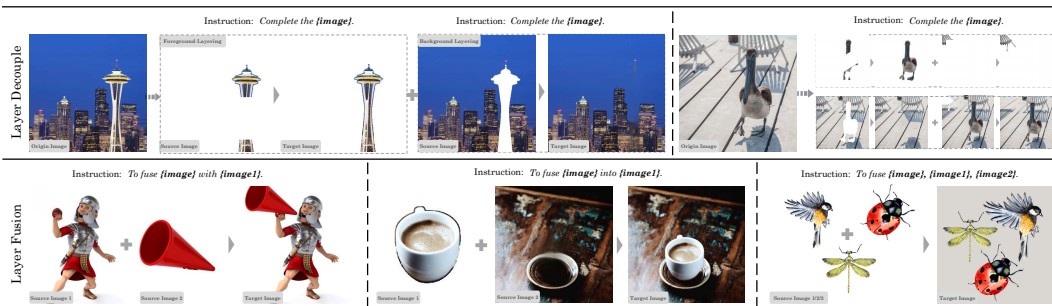

Figure 21: Sample data for training layer decouple and layer fusion in layer editing.

requirements. In constructing this dataset, we not only employ random masks paired with corresponding textual descriptions of the original images but also refine the process to focus on local regions. First, we obtain multiple object masks from the image, and then extract detailed textual descriptions for each object. Finally, we create triplets consisting of the original image, the local object mask, and the local object caption. This approach enables the generation of richer and more controllable details within local areas.

### B.6.3 OUTPAINTING

The outpainting task involves intelligently generating and completing the edge regions of an existing image so that the extended new image appears natural and continuous visually. The major challenge of this task is producing images that are rich in detail, diverse in content, and exhibit a certain level of associative ability. In terms of data processing, we employed commonly used techniques, applying random masks to the areas and directions that need to be expanded, in order to adapt to different scenarios of image completion.

### B.7 LAYER EDITING

Hierarchical layer editing operations on images involve two aspects: **(i) Layer decouple:** enables the separation of the main subject within a single image, resulting in a complete subject and a reconstructed background. The subject must be restored to its complete form, mitigating any gaps caused by occlusion or other reasons present in the original image. Meanwhile, the background is filled in for the blank areas left after the subject's separation, achieving a fully deconstructed fore/background. **(ii) Layer fusion:** allows for the incorporation of distinct independent subjects into a target image, facilitating high-quality image integration. The inference pipeline can be seen in Fig. 20.

For the data construction, we follow the data workflow from the object editing task, focusing on slightly larger subjects and data containing multiple subjects within a single image. This approach

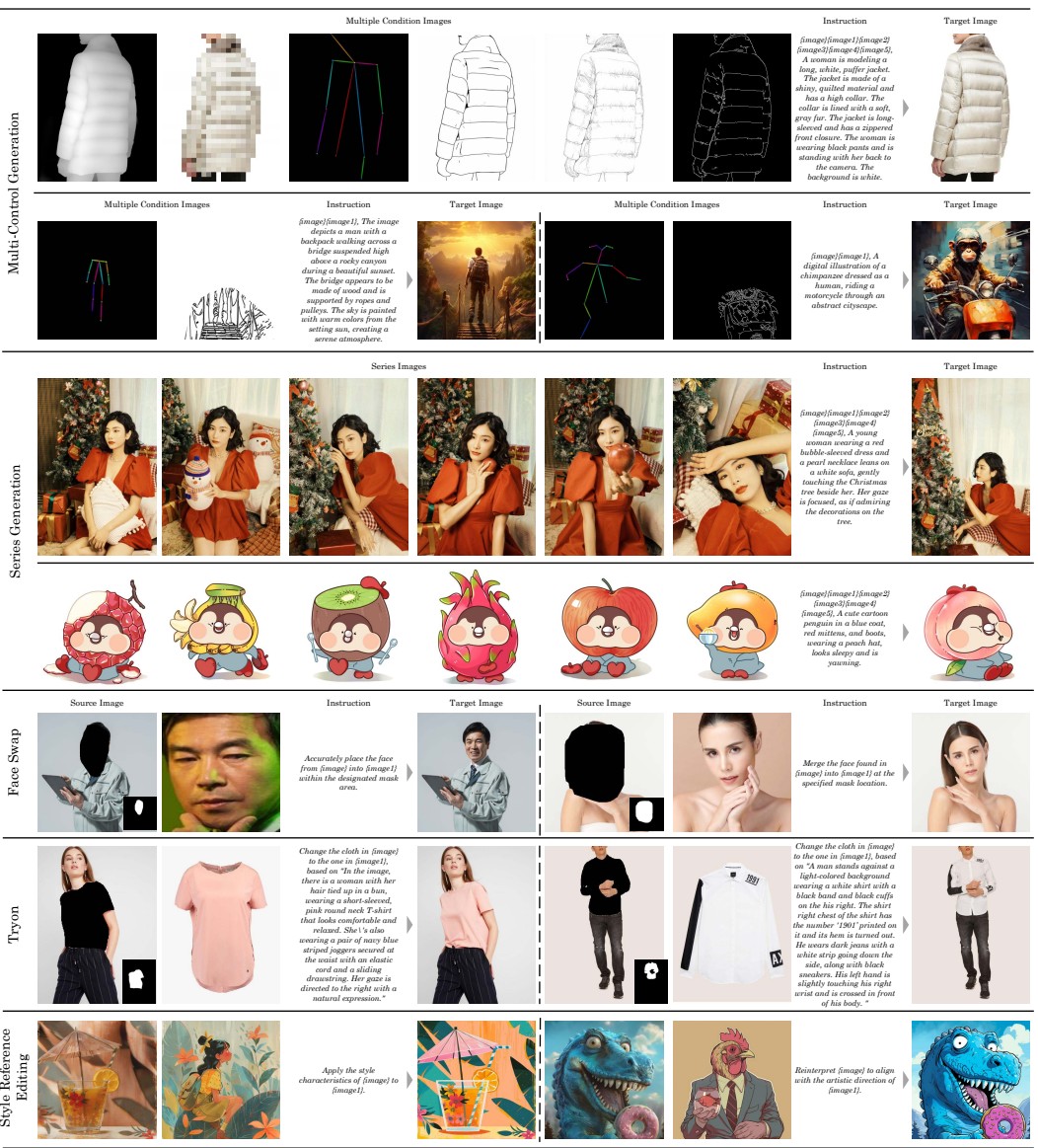

Figure 22: The dataset visualization of multi-reference.

creates compositions that allow for a lossless splitting of a single original image into multiple sub-images, and conversely establishes a correspondence for combining several images into one. Specifically, as shown in Fig. 21, in layer decouple stage, we follow the instructions to transition from the original image to either a singular subject or a singular background. During training, the non-subject areas of the subject image and the incomplete portions of the background are filled with white color. Additionally, to simulate the scenario of subjects obscured in the image, we perform random masking on the extracted subject images. The output targets are the complete subject or background. In layer fusion stage, we employ a multi-reference image strategy, taking single or multiple subjects along with the background as inputs to guide the generation of the target image. Similarly, different subjects are supplemented with white color and placed on a randomly sized white canvas, with the training goal being to generate a harmonious and complete composite image.

## B.8 REFERENCE GENERATION

Ordinary image generation and editing tasks require no more than one input image. Under certain circumstances, image generation needs multiple image inputs, such as multiple conditions in con-

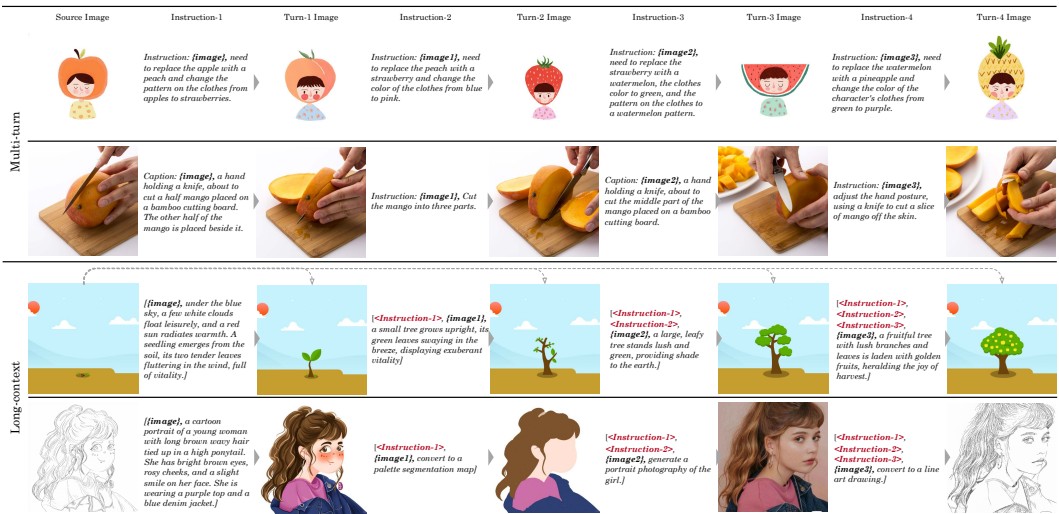

Figure 23: Sample data for training multi-turn and long-context generation task.

trollable generation, and a group of character design images for ID preservation. The same is true for editing tasks, one or more additional exemplar images are necessary to specify the expected visual elements in the editing area. For example, a reference image can be interpreted as the target image style appearance, face identity, *etc*. Therefore, we prepare training data for multi-reference generation and reference-guided editing. Examples of training data are shown in Fig. 22.

### B.8.1 MULTI-REFERENCE GENERATION

**Multi-Condition Generation.** In controllable generation, overlaying different types of conditions is usually necessary to control the different visual aspects of generated images. Similar to the process in appendix B.3, canny edge maps, depth maps, color maps, grayscale images, contours, scribbles, doodles, and pose keypoints are included for multi-condition generation. To make it possible to composite objects in different conditions, we use object segmentation to assign each area with a different condition.

**Series Generation.** It has been widely studied how to generate images about one consistent visual element, like the portrait of a specific figure, pictures with the same styles, *etc*. Usually, tuning a themed tuner (*e.g.*, LoRA) (Hu et al., 2022) with few images is the primary option. However, we are aiming to teach our model to understand and follow the rules lying behind image series. We collect image groups through image clustering. During the training phase, we randomly sample one image in the cluster as a target and 3 to 8 images as input images.

### B.8.2 REFERENCE-GUIDED EDITING

Style and face are two typical editing tasks benefiting from additional reference image inputs, providing supplementary visual information of the target images. Virtual Try-on is an editing task naturally requiring a cloth image as reference to edit the person image. We establish a large training data for these 3 tasks.

**Style Reference Editing.** To construct the training data, we extend the data of style editing (appendix B.4.2) by assigning an additional style reference image for each edit-target image pair. Reference images are randomly selected from other styled images within the same style category.

**Face Reference Editing.** We use image pairs of misaligned facial data (appendix B.4.1) for face reference editing. We pick one of the two images as reference image while another as target image. Therefore, the target and reference image are the same person but slightly different. The edit image is derived from target image by erasing the face area to avoid any spoilers.

**Virtual Try-on.** We use public high resolution datasets VITON-HD(Choi et al., 2021) and Dress Code(Morelli et al., 2022) as training data, with over 100k records in total.

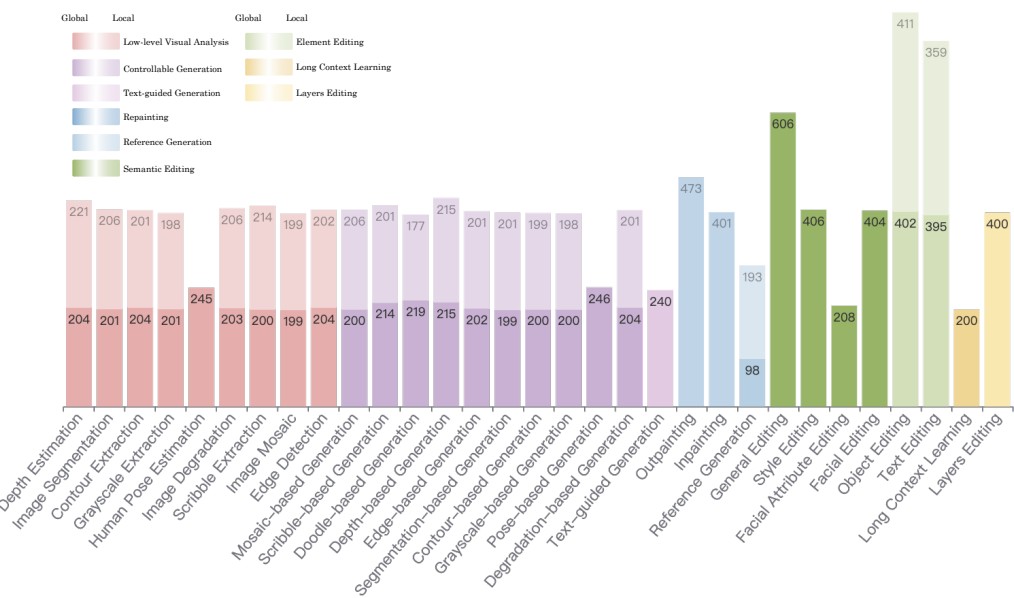

Figure 24: **The overview of benchmark distribution.** "Global" and "Local" refer to editing or generating based on the entire image, and editing or generating based on specific local areas of the image, respectively.

Table 3: The comparison between ACE benchmark and existing benchmarks.

| Benchmark | Real Image? | Generated Image? | Multi-turn? | Regional? | Tasks | Data Scale |
|---|---|---|---|---|---|---|
| MagicBrush | Y | N | Y | Y | - | 1588 |
| Emu Edit | Y | N | N | N | 8 | 3589 |
| ACE | Y | Y | Y | Y | 31 | 12000 |

## B.9 MULTI-TURN AND LONG-CONTEXT GENERATION

Multi-turn editing refers to the process of obtaining the final image from an input image through multiple independent instruction-based editing, which poses significant challenges in both the model's precise understanding of instructions and the control over image quality in every round. Further, the long-context generation process aims to leverage the contextual information provided in each round of interactions to construct a long sequence, thereby generating images that align with the intended directives. The generated images reference multiple images and their corresponding instruction information from previous interactions, capturing the user's genuine intent within the interaction framework, and allowing for more precise image editing. The sample data can be seen in Fig. 23.

The data construction consists of two parts: (i) Homogenous content-based condition unit: this involves employing a pair data collection strategy to obtain various clusters from a large-scale database, as shown in Sec. 3.1, where each cluster contains images paired with their respective captions and instruction generated in pairs. During training, we select one image from any chosen cluster as the starting point and build a multiple rounds data chain using its caption or instruction, predicting the final image as the endpoint of the chain. (ii) Task-based condition unit: we treat all the previously mentioned single-image tasks as individual turns within the task and randomly sample them to form a complete multiple precursor unit that guides the final image generation.

## C BENCHMARK DETAILS

Previous methods have proposed benchmarks to evaluate model performance for image editing, with notable examples including MagicBrush (Zhang et al., 2023a) and Emu Edit (Sheynin et al., 2024).

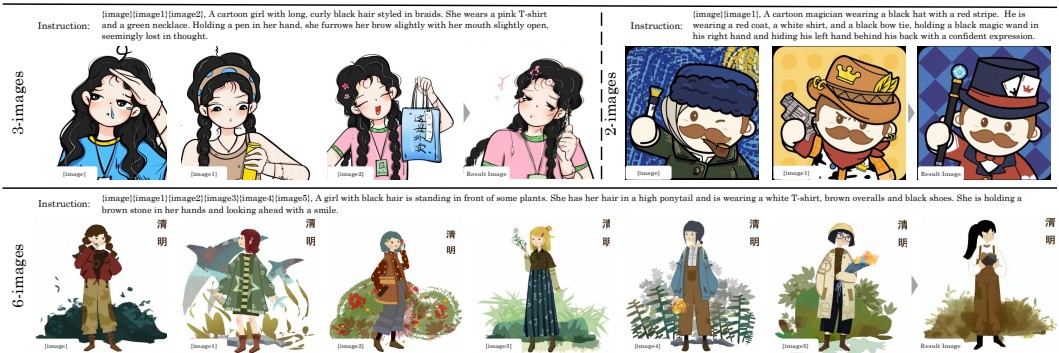

Figure 25: **The effectiveness of Long-context Attention Block.** Benefit from this, our model is capable of accepting image sequences of varying lengths as input while generating coherent and reasonable outputs.

MagicBrush has 1,588 samples, which includes 1,053 single-turn and 535 multi-turn instances, and primarily comes from MS-COCO (Lin et al., 2014). Emu Edit first defines 8 different categories of potential image editing operations and constructs the instructions by human annotators. The main issues with the above methods are insufficient coverage of tasks and generally poor data quality. ACE builds a benchmark comprising 12,000 samples, covering more than 31 tasks while accommodating 5900 real images and 6100 generated images. In addition, ACE benchmark supports both regional editing and multi-turn editing tasks. The specific statistics are shown in the Fig. 24, and the comparison with other benchmarks is presented in the Tab. 3.

# D ARCHITECTURE DESIGN

We employ the Diffusion Transformer (DiT) Peebles & Xie (2023) as our foundational architecture. However, two significant challenges arise during training for editing tasks: (i) *How can we effectively manage input image sequences of varying lengths?* and (ii) *How can we accurately align an image sequence with multiple images as specified by the accompanying natural language instructions?* To tackle these challenges, we propose the **Long-context Attention Block** and **Image Indicator Embeddings**, which will be elaborated upon in this section.

**Effectiveness of Long-context Attention Block.** Our objective is to develop an all-round foundational model for various editing tasks. However, the standard DiT architecture struggles with managing image sequences of varying lengths. Conventional editing methods tend to focus on a limited set of editing tasks and typically concatenate images along the channel dimension, which is not suitable for our model. To address this limitation, we propose the Long-context Attention Block, which concatenates images along the token dimension, thereby enabling support for flexible input shapes. In Fig. 25, we present samples featuring different numbers of input images alongside the output images generated by our proposed method. The results demonstrate that our model can effectively accommodate a variety of input image sequences while producing coherent and contextually appropriate outputs.

**Effectiveness of Image Indicator Embeddings.** We evaluate the effectiveness of Image Indicator Embeddings (IIE) by manipulating the sequence of input images. As illustrated in Fig. 26 and Fig. 27, models trained with IIE consistently succeed in accurately linking the correct image to its corresponding textual mention. In contrast, models trained without IIE frequently struggle to make these associations. For instance, in the left case illustrated in Fig. 27, the edit image is represented by {image}, while the style reference image is denoted as {image1}. The model trained with the IIE can successfully generate the desired edit operation, producing an image that retains the content of {image} while adopting the style of {image1}. In contrast, the model trained without IIE tends to misinterpret the semantic context, leading to an erroneous editing operation that modifies the style reference image instead. This suggests that Image Indicator Embeddings facilitate an semantic connection between textual instructions and their associated images.

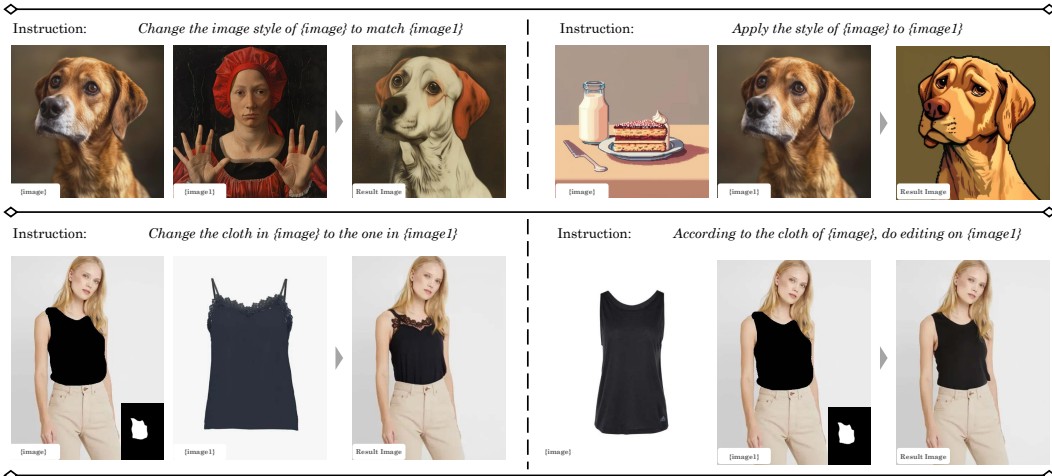

Figure 26: **The effectiveness of Image Indicator Embeddings (IIE).** Model trained with IIE succeed in accurately linking the correct image to its corresponding textual mention.

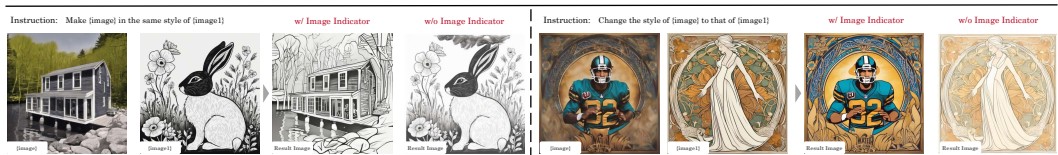

Figure 27: **The comparison between models trained with and without Image Indicator Embeddings (IIE).** Model without IIE frequently confuses edit images (denoted as {image}) with reference images (designated as {image1}). In contrast, model trained with IIE demonstrates the ability to accurately associate images with their corresponding indicator tokens.

# E    IMPLEMENTATION DETAILS

**Data Preprocessing Details.** The proposed hierarchical aggregation pipeline for pairing content-related images from extensive databases involves three key steps: first, clustering based on SigLip features; second, identifying first-level disjoint sets using the similarity of SigLip features; and third, determining second-level disjoint sets by examining task-specific correlations. The methodological details are described as follows:

To enhance efficiency in executing the Union-Find algorithm, particularly since calculating similarities at the billion-level would require over 1 PB of RAM, we initially cluster data based on SigLip features. This clustering serves as a preprocessing step for identifying the first-level disjoint sets derived from SigLip feature similarities. To manage data scale effectively and keep its scale under 100K, we apply the K-means algorithm to create 10,000 clusters. This approach facilitates the parallel execution of the Union-Find algorithm using approximately 128 V100 GPUs.

To create higher correlation groups compared to the clusters formed in the previous step, we utilize first-level disjoint sets based on the similarity of SigLip features. Within a specific data cluster, we calculate the correlation between pairs of samples to build a data graph using the SigLip similarity matrix. We set an upper threshold of 0.98 and a lower threshold of 0.85 for data pruning. Upper threshold is mainly used for removing duplicate data. By applying the Union-find algorithm, we can identify a cohesive data group. The threshold range of 0.85 to 0.98 ensures a strong connection among the data within each group.

Identifying second-level disjoint sets based on task-specific correlations predominantly involves advanced data mining tailored to the various definitions of data correlation required for different tasks. For instance, in editing tasks like scene or background alterations—where the image's main subject should remain unchanged while only the background is modified—it's essential to train a specialized correlation model. This model ensures that only the background changes in the paired data. We train

Table 4: The hyperparameters setting for task-specific disjoint sets mining.

| Tasks | Criteria Method | lower threshold | higher threshold |
|---|---|---|---|
| General Editing | Binary Classification | 0.9 | 1.0 |
| Misaligned Facial Editing | In-house Facial Feature Extractor | 0.8 | 0.9 |
| Object Editing | Binary Classification | 0.5 | - |
| Text Editing | Binary Classification | - | 0.99 |
| Reference Generation | Binary Classification | 0.8 | 0.95 |

Table 5: The multi-stage training details for ACE.

| Stage | Model Capacity | Train Data Scale | Visual Sequence Length | Max Image Number | Training Steps | Batch Size |
|---|---|---|---|---|---|---|
| Instruction Align | 0.6B | 0.7 Billion | 1024 | 1 | 900K | 800 |
| Instruction Align | 0.6B | 0.7 Billion | 1024 | 9 | 100K | 400 |
| Aesthetic Improvement | 0.6B | 50 million | 1024 | 9 | 500K | 400 |
| Aesthetic Improvement | 0.6B | 50 million | 4096 | 9 | 100K | 960 |

this correlation model using the ViT-B-16-SigLIP as the backbone for binary classification, setting the lower threshold of 0.96 to maintain control over the correlation. For tasks involving portrait ID preservation, the key requirement is to maintain consistency in the identity (ID) of paired data. In such cases, the correlation is measured using the cosine distance of facial features, with the threshold ranging of 0.8 to 0.9 to ensure correlation. We list the specific correlation measures applicable to different tasks in Tab. 4.

**Training Details.** We employ the T5 language model as the text encoder and DiT-XL/2 (Peebles & Xie, 2023) as the base network architecture. The model capacity is nearly 0.6B and the parameters are partly initialized by PixArt-$\alpha$ (Chen et al., 2023a). The maximum length of the text token sequence is set to 120. We freeze VAE and T5 modules, utilizing AdamW (Loshchilov & Hutter, 2018) optimizer to train the DiT module with a weight decay of 5e-4 and a learning rate of 2e-5. All experiments are conducted in A800.

A multi-stage training strategy is employed to progressively enhance the aesthetic quality and increase the generalizability of model. The training details are presented in Tab. 5. First, we train the instruction-following capability on single-image tasks using 0.7 billion data points using 80 A100 cards, with the number of single image tokens limited to 1024. Gradient checkpoint and data parallel technologies are utilized to ensure a sufficient batch size. Next, we expand the tasks to include multiple-image scenarios. After learning the instruction alignment, we utilize high-aesthetic data to enhance the model's aesthetics and extend the max image token number to 4096 to generate higher resolution image.

**Computational Efficiency.** The training speed is influenced by the visual sequence length of each image and the number of input images, as shown in Fig. 28a. When using a visual sequence length of 1024 and only training with single image editing and text-to-image tasks, which involve no more than 1 input image, ACE can be trained at a speed of approximately 198,000 records per GPU day on average, using the A800-80G GPU. Incorporating a fraction of multi-image tasks and setting the maximum number of input images to 9, the training speed declines slightly to approximately 181,000 records per GPU day with the same GPU setup. If the visual sequence length is increased to 4096, the average throughput drops significantly to around 51,000 records per GPU day. During inference, we set the sample steps to 20 and test the single image editing inference time cost. The average time cost is 3.46s for visual sequence length 1024 and 11.18s for visual sequence length 4096.

**Checkpoints Evaluation.** To gain a deeper understanding of the impact of data scale, we evaluate the intermediate model checkpoints at different training steps on the MagicBrush benchmark. Each of them are trained with different amount of data. In Fig. 28b, We present the relative score changes comparing with the score of 15k step checkpoint. A clear trend of performance improvement is observed in this evaluation. In general, the model benefits from being trained with more data.

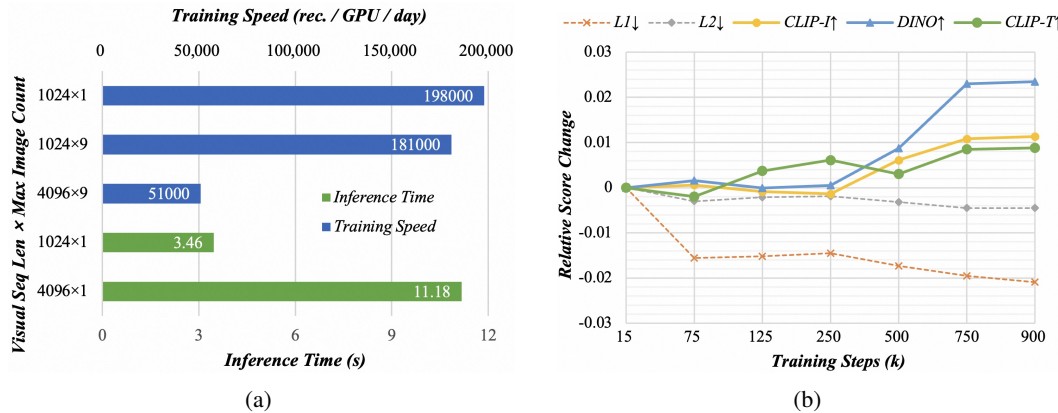

Figure 28: **Computational Efficiency and Checkpoints Evaluation.** a) Training speed and inference time cost under different settings. b) Changes in relative scores of MagicBrush benchmark metrics. Taking the model at 15k step as baseline, the scores generally improve with an increase in training steps. The x-axis is plotted with non-uniform intervals.

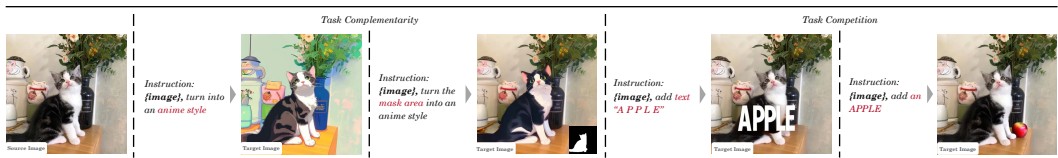

Figure 29: **Complementarity and competition between tasks.** The complementarity of tasks is exemplified by the capability of style transfer for local editing; the competition between tasks is reflected in the potential for confusion in task execution due to ambiguous instructions.

**Task Interdependence.** In an all-in-one visual generation model, there exist multiple interactions between tasks, similar to that in large language models, and this relationship can be viewed as a complex balancing action. **(i) Complementarity between tasks**: The combined influence of various tasks can lead to a certain degree of generalized behavior across tasks. For instance, in the style transfer task, our prepared data and training process focus on pixel-aligned global image transfer. However, by incorporating learnings from other tasks related to mask guidance or subject guidance, the model can acquire the ability to perform style transfer in localized areas. **(ii) Competition between tasks**: As the scale of tasks increases, the potential for competition also grows, particularly in scenarios where user instructions are ambiguous. For example, when adding the text 'APPLE' to an image, it is essential to specify that it is text to be added; otherwise, due to semantic ambiguity, the result may instead involve the addition of an object depicting an apple.

To achieve optimal performance balance, we first focus on adjusting the data sampling rates for each task in a phased manner during the training process, monitoring this through a validation set. Additionally, more detailed descriptions of instructions are needed in the preparation of training data to prevent semantic confusion between tasks. Through these methods, we aim to ensure that the model can fully leverage the complementarity between different tasks while controlling for any potential negative impacts. However, the relationships between different tasks still require further exploration to better optimize the model's performance. Future work will also focus on how to effectively evaluate and adjust these influencing factors to achieve a more balanced and comprehensive execution of tasks.

**Visualization of Editing Process.** We visualize a number of intermediate model outputs of the denoising process in Fig. 30. At the initial steps, the model selects an appropriate region within the input image for editing based on the textual instructions. In the subsequent steps, the unchanged region is entirely replicated from the input image. Following this, the edited area gradually accumulates details and begins to reach completion.

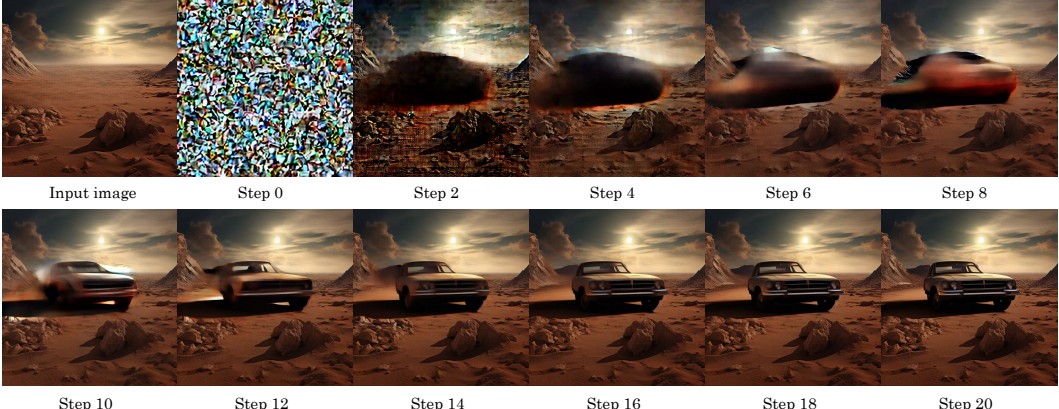

| | | | | | |
|---|---|---|---|---|---|
| Input image | Step 0 | Step 2 | Step 4 | Step 6 | Step 8 |
| Step 10 | Step 12 | Step 14 | Step 16 | Step 18 | Step 20 |

Figure 30: **Visualization of editing process.** The model identifies the area to be edited in the initial steps and subsequently copies the unchanged regions from the input image in the following steps. In the steps leading up to the final stage, additional details are incrementally added to the edited area until completion. The instruction is "Let a car appear in {image}."

Table 6: Results on Emu Edit benchmark. ACE shows comparable performance to its baselines.

| Method | CLIPdir↑ | CLIPout↑ | L1↓ | CLIPimg↑ | DINO↑ |
|---|---|---|---|---|---|
| InstructPix2Pix (Brooks et al., 2023) | 0.0739 | 0.2681 | 0.1240 | 0.8508 | 0.7647 |
| MagicBrush (Zhang et al., 2023a) | 0.0831 | 0.2701 | 0.0995 | 0.8664 | 0.7927 |
| Emu Edit (Sheynin et al., 2024) | **0.1073** | **0.2791** | 0.0893 | 0.8743 | 0.8398 |
| UltraEdit (Zhao et al., 2024) | 0.0888 | 0.2783 | **0.0532** | 0.8814 | 0.8524 |
| CosXL (StabilityAI, 2024) | 0.0901 | 0.2775 | 0.0940 | 0.8686 | 0.8340 |
| **ACE** (Ours) | 0.0855 | 0.2746 | 0.0761 | **0.8952** | **0.8620** |

## F    MORE EXPERIMENTS

**Emu Edit Benchmark.** We also conduct a comparison on Emu Edit benchmark (Sheynin et al., 2024). It includes 3,589 examples of 8 tasks: background alteration, comprehensive image changes, style alteration, object removal, object addition, localized modifications, color/texture alterations, and text editing. This benchmark measures the similarity between output and input images and the provided captions. We calculate the L1 distance, CLIP similarity, and DINO similarity between the generated image and input image, together with the CLIP text-image direction similarity measuring agreement between the change in captions and the change in images, and CLIP similarity between the generated image and output caption. We use the code adapted from the MagicBrush evaluation code and models of CLIP ViT-B/32 and DINO ViT-S/16. As shown in Tab. 6, ACE achieves comparable performance to its baselines.

**Facial Editing**. When evaluating the face identity preservation ability, we designed a Face Similarity (FS) metric to measure the consistency of faces between generated images and original images. We first detect the face region using MTCNN (Zhang et al., 2016b), then extract face embeddings with the ArcFace (Deng et al., 2019a) algorithm. The cosine similarity between normalized embeddings is calculated as the face similarity score. The images generated by MagicBrush and InstructPix2Pix exhibit excessive similarity to the original images, thus metrics for these two methods are not computed. We observed a non-linear growth in the Face Similarity score. To analyze this, we extracted facial features from over 5 million data points in MS1M_V3 (Deng et al., 2019b) and grouped them into clusters based on their face_id. We then calculated the pairwise similarity within each cluster, resulting in a mean of $mean = 0.5258$ and a standard deviation of $std = 0.1765$. Due to the large standard deviation, we further analyzed the percentage of samples with scores above $0.3493(mean - std)$ that met the instructions to evaluate the Effective Score(ES) of facial ID persistence.

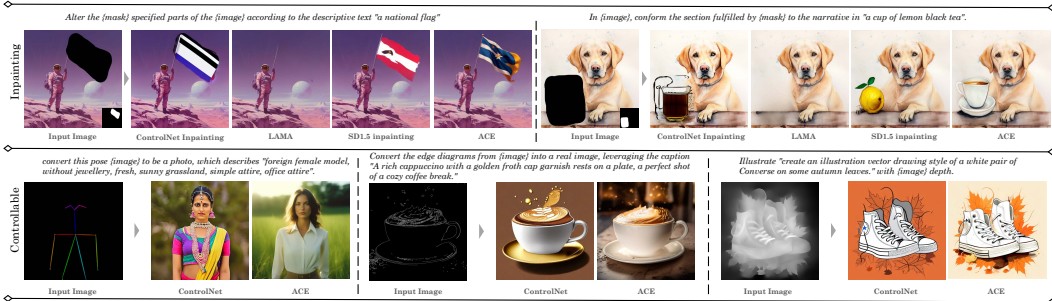

Figure 31: **More Qualitative Comparison with SOTA Methods.** We compare ACE with more baselines of inpainting and controllable generation tasks.

Table 7: Quantitative results for Facial Editing tasks on ACE benchmark. † indicates that InstanID requires an additional landmark as a condition, while other methods do not.

| Method | Face Similarity | Effective Score |
|---|---|---|
| InstantID[†] Wang et al. (2024b) | 84.08 | 0.96 |
| CosXL StabilityAI (2024) | 66.49 | 0.37 |
| UltraEdit Zhao et al. (2024) | 62.91 | 0.16 |
| IP-Adapter Ye et al. (2023) | 66.51 | 0.31 |
| FaceChain Liu et al. (2023b) | 65.46 | 0.42 |
| **ACE** (Ours) | **70.07** | **0.67** |

Table 8: Quantitative results for Local Text Render tasks on ACE benchmark.

| Method | Edit Distance | Sentence Accuracy |
|---|---|---|
| UDiffText (Zhao & Lian, 2024) | 0.6827 | 0.4110 |
| AnyText (Tuo et al., 2023) | 0.6035 | 0.3313 |
| **ACE** (Ours) | **0.8211** | **0.5767** |

Our model significantly outperforms other methods in the absence of facial landmark information, with improvements of **3.56%** and **25%** in the FS and ES metrics as shown in Tab. 7. Although our model (0.6B) demonstrates inferior performance on metrics compared to InstantID (2.6B), it is important to highlight that InstantID utilizes an additional facial landmark as a conditioning factor. Moreover, as indicated by the results of prompt-following and image quality assessments in Tab. 2, our model shows a highly competitive performance overall.

**Local Text Render**. To adequately evaluate the performance of text editing, we provide the quantitative analysis of our method with two SOTA text render methods, *i.e.*, UDiffText, and Anytext, on the local text render task of ACE benchmark. Each generated text line is cropped according to the specific position and fed into an OCR model to obtain predicted results. As described in Anytext, we calculate the Sentence Accuracy and the Normalized Edit Distance for each method. The former metric evaluates the sentence-level accuracy and the latter metric evaluates the char-level precision. From Tab. 8 we can observe that our ACE outperforms the other two methods, achieving performance gains of **14%** and **16%** in terms of Normalized Edit Distance and Sentence Accuracy, respectively. This demonstrates our superior text rendering capability.

**More Qualitative Comparison**. We compare ACE with specifically designed state-of-the-art methods of inpainting and controllable generation tasks in Fig. 31. For the inpainting task, ControlNet Inpainting frequently arrives an output with black artifacts, incapable of inpainting an image with a black mask perfectly. LaMa always interprets inpainting as a removal task, as it doesn't receive any text as condition inputs. Broadly, in inpainting and controllable generation tasks, ACE shows a significant advantage in terms of aesthetics and text prompt following ability compared to SD1.5 inpainting and ControlNet.

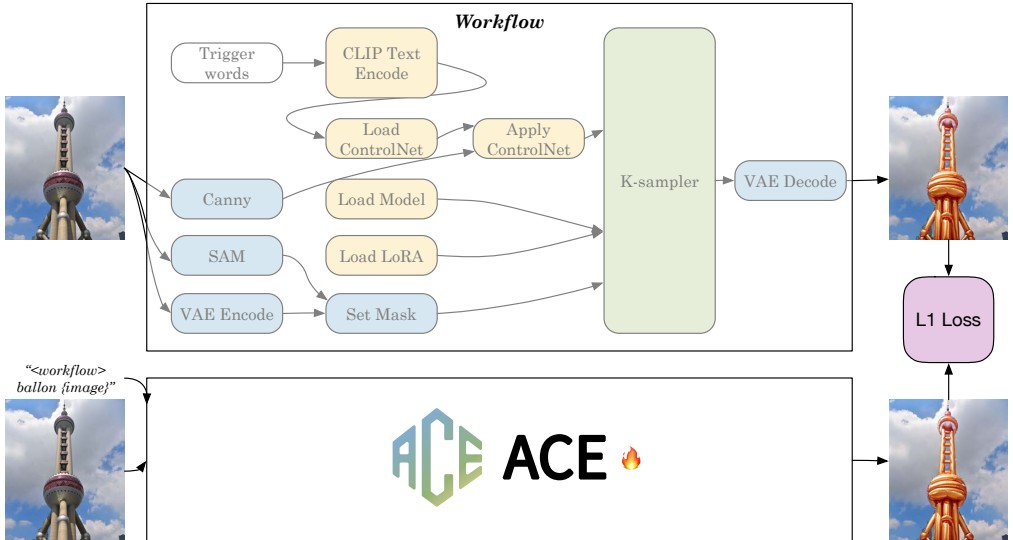

Figure 32: The pipeline of workflow distillation.

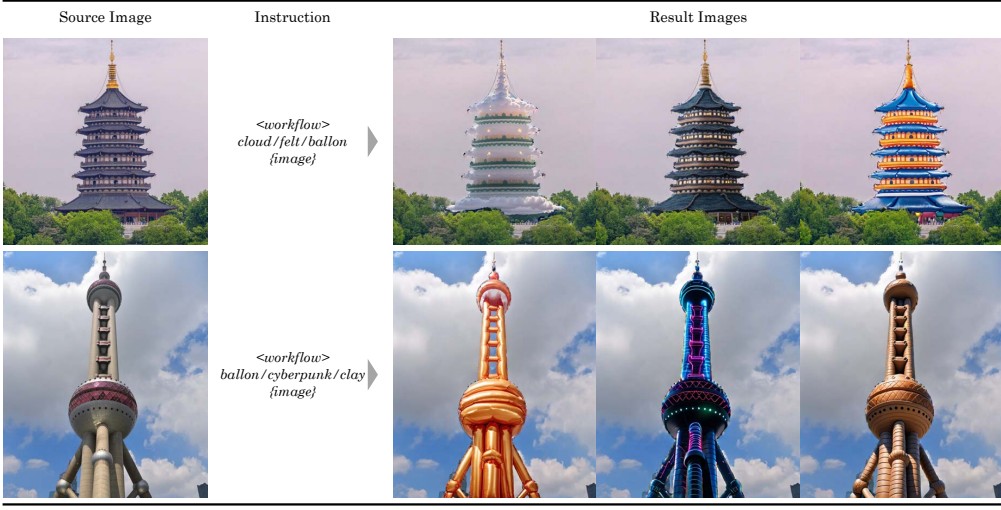

Figure 33: The results visualization of workflow distillation.

# G APPLICATION

## G.1 WORKFLOW DISTILLATION

There are many excellent workflows assembling LoRAs, ControlNets, and T2I models on open-source platforms, which enable users to achieve certain results. To show the capability and compatibility of ACE, we collect several outstanding workflows to obtain their result images for distillation. We train ACE with the inputs and corresponding outputs of these workflows, as well as a fixed special trigger instruction, as illustrated at Fig. 32. Our model acquires similar abilities of these workflows, as shown in Fig. 33, which demonstrates the great potential of ACE.

## G.2 CHAT BOT

Leveraging our diffusion model, we build a chat bot application to achieve chat-based image generation and editing. Rather than a cumbersome visual agent pipeline, our chat bot supports all image creation requests with only one model serving as the backend, hence achieving significant efficiency

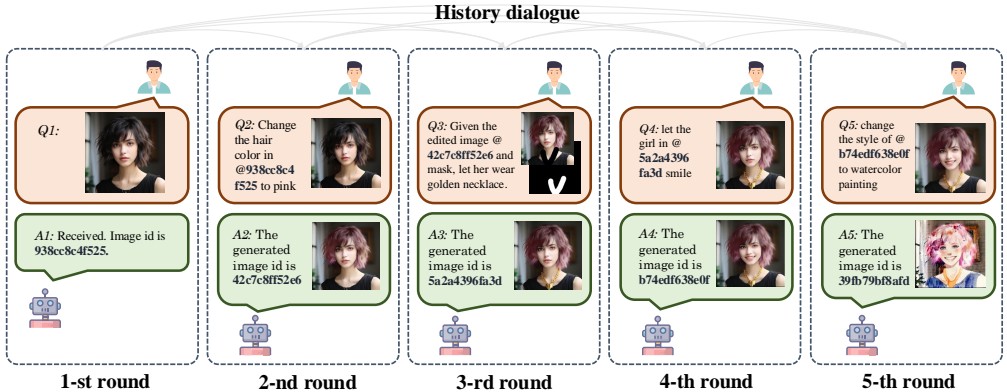

Figure 34: The multi-turn conversation pipeline of our chat bot application.

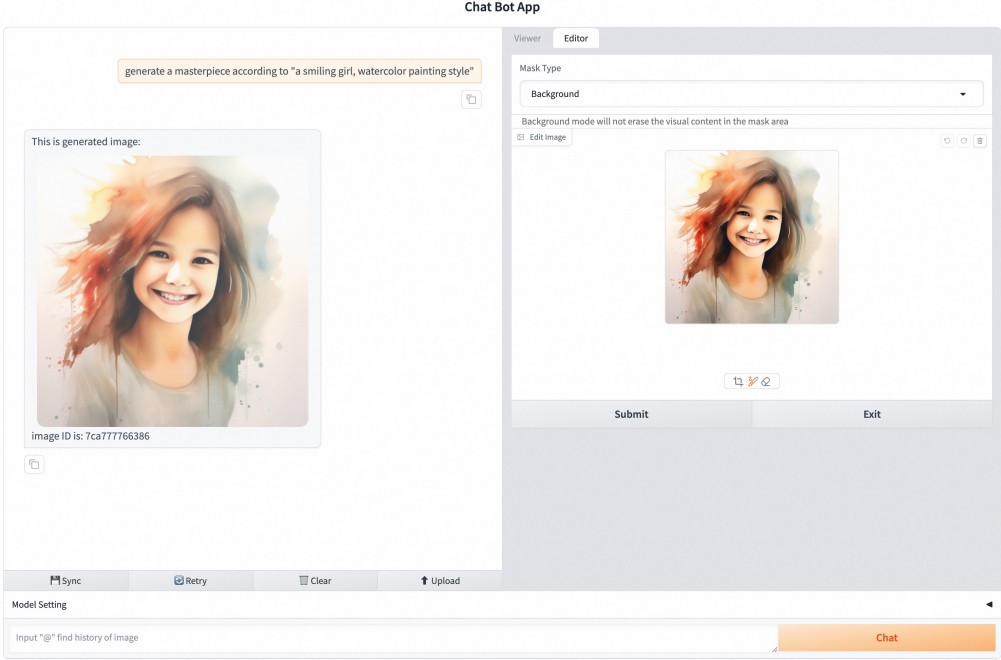

Figure 35: The user interface of the chat bot application built with Gradio.

improvement compared with visual agents. We depict a multi-turn conversation sample in 34 and illustrate the user interface in Fig. 35. We could command the model to create any desired image by chatting with it using natural language. The overall system can be formulated as

$$A_j = ChatBot(H_{<j}, Q_j), \tag{6}$$

where $A_j$ denotes the $j$-th round output of chat bot, $Q_j$ represents the $j$-th round user request, and $H_{<j} = \{(Q_1, A_1), (Q_2, A_2), ..., (Q_{j-1}, A_{j-1})\}$ represents the history of dialogue before $j$-th round. By introducing the history dialogue information into the current conversation, our model excels at understanding complex user requests, therefore achieving better prompt following ability.

## H  MORE VISUALIZATION

In Fig. 36, and Fig. 37, we present the visualization results of ACE in low-level visual analysis. The Fig. 38, Fig. 39, and Fig. 40 are the visualization of controllable generation. The visualization results of repainting are depicted in Fig. 41. Semantic editing tasks such as general editing, facial

editing, and style editing are illustrated in Fig. 42, Fig. 43, Fig. 44. The visualization results of elements editing including text editing, and object editing are shown in Fig. 45, and Fig. 46. In Fig. 47, we present the visualization results of layer decouple and layer fusion. The visualization of reference generation can be found at Fig. 48 and that of multi-turn and long-context generation are present in Fig. 49. ACE demonstrates proficient instruction following, high-quality generation, and versatility across different tasks.

## I  DISCUSSION

**Societal Impacts.**

From a positive perspective, the intelligent generation and editing of images can provide artists and designers with innovative tools to inspire new concepts, enhance creativity and artistic expression in images, lower the barriers to artistic creation, and reduce the labor-intensive manual processes involved. Additionally, the method can serve various industries. In the field of education and training, they can be used to create supplementary teaching materials, such as illustrations for picture books, enhancing students' learning experiences and improving communication and understanding in lessons. In business environments, companies can utilize the method to generate marketing materials and product designs, thereby increasing production efficiency and creative output. The positive impacts of these technologies offer new possibilities for creativity, educational quality, and business efficiency, making it worthwhile for us to actively explore and apply them.

New technologies not only bring new opportunities but also come with challenges. Firstly, issues related to copyright and authorship are prominent, potentially infringing upon the rights of original works and leading to legal disputes. Secondly, false information generated by models may exacerbate the spread of rumors, undermining public trust in information. Lastly, the inherent biases and stereotypes present in generated content can challenge societal values and moral standards. Therefore, while we acknowledge the conveniences and innovations offered by these technologies, it is imperative to carefully consider and effectively manage these negative impacts to ensure the sustainability of technological development and uphold social responsibility.

**Limitations.**

First, our approach offers a unified framework for existing editing tasks. For specific tasks such as text-to-image generation, the aesthetic quality of our generated results lags behind that of state-of-the-art generative models like Midjourney and FLUX. These models have achieved breakthroughs by focusing on a single task of generating images from text prompts. In contrast, our model supports a broader range of input types and handles a wider variety of tasks, such as performing diverse edits under open-ended instructions. Additionally, training on higher-quality data and using a larger-scale model could help bridge this gap.

Second, the model for instruction editing needs to accurately capture the user's actual intent. In our framework, we utilize a fixed encoder-decoder language model to encode text instructions. However, as user instructions become more complex and diverse, the difficulty of interpreting these instructions also increases. Furthermore, the current model is unable to handle multiple intents or tasks from a single instruction simultaneously and requires intent decomposition.

Third, we support the input of multiple images and multiple instructions to construct long contextual information for generation. On one hand, it is inevitable that, due to limited hardware resources, training and inference with multiple images become increasingly challenging as the number of tokens increases. On the other hand, excessively long contextual inputs pose a significant challenge for the model, as the forgetting of historical information during the process may lead to biases in the final generated results.

**Future Work.**

We try to illustrate some existing constraints of the model in limitations, which can serve as directions for our future work. Firstly, the phenomenon of scaling laws has been demonstrated in the NLP field, indicating that further exploration of scaling laws in complex generation tasks is warranted. We will focus on two main approaches: on one hand, we will work on expanding high-quality data, which includes improving data quality, incorporating more complex tasks, and enhancing the precision of instructional data; on the other hand, we will directly increase the model architecture's scale

to enhance its general generative capabilities. Secondly, we aim to introduce LLMs or MLLMs to accurately capture the intentions of users at the instruction level, leveraging their robust general understanding of language and images. This involves two specific objectives: firstly, to enhance the model's ability to generalize from input text instructions to match single-task or multi-task contexts; and secondly, to improve the understanding of input images that are to be edited, thereby assisting subsequent instruction operations and ensuring the accuracy and diversity of the generated content. Finally, it is essential to explore the long-sequence modeling of multi-modal data comprising multiple rounds of image and text interactions. We will engage in continuous contemplation regarding how to ensure that the historical context of image-text pairs is truly beneficial, similar to the functionality of chatGPT-like language models.

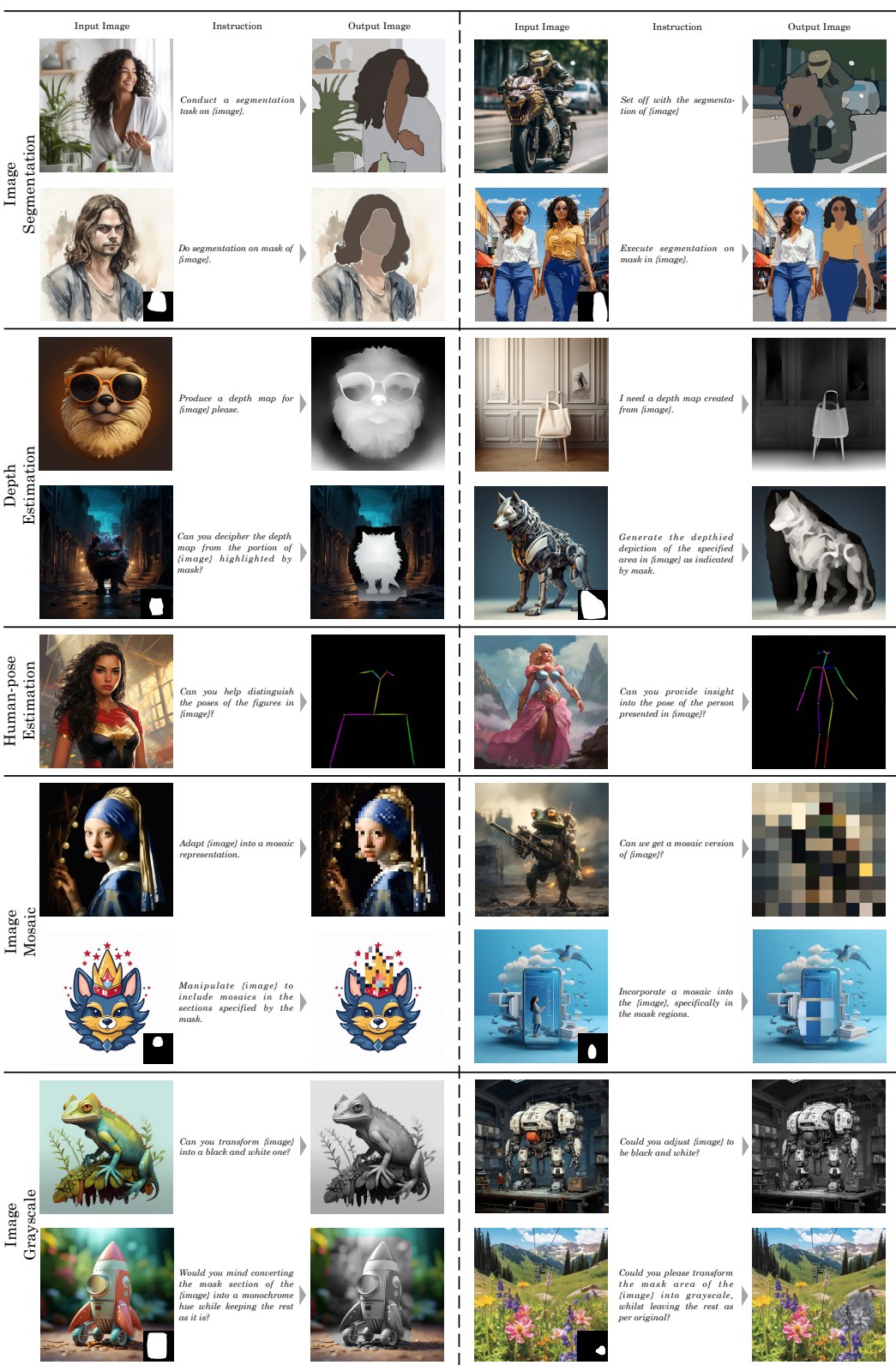

Figure 36: The ACE's generated visualization of image segmentation, depth estimation, human-pose estimation, image mosaic, and image grayscale in low-level visual analysis.

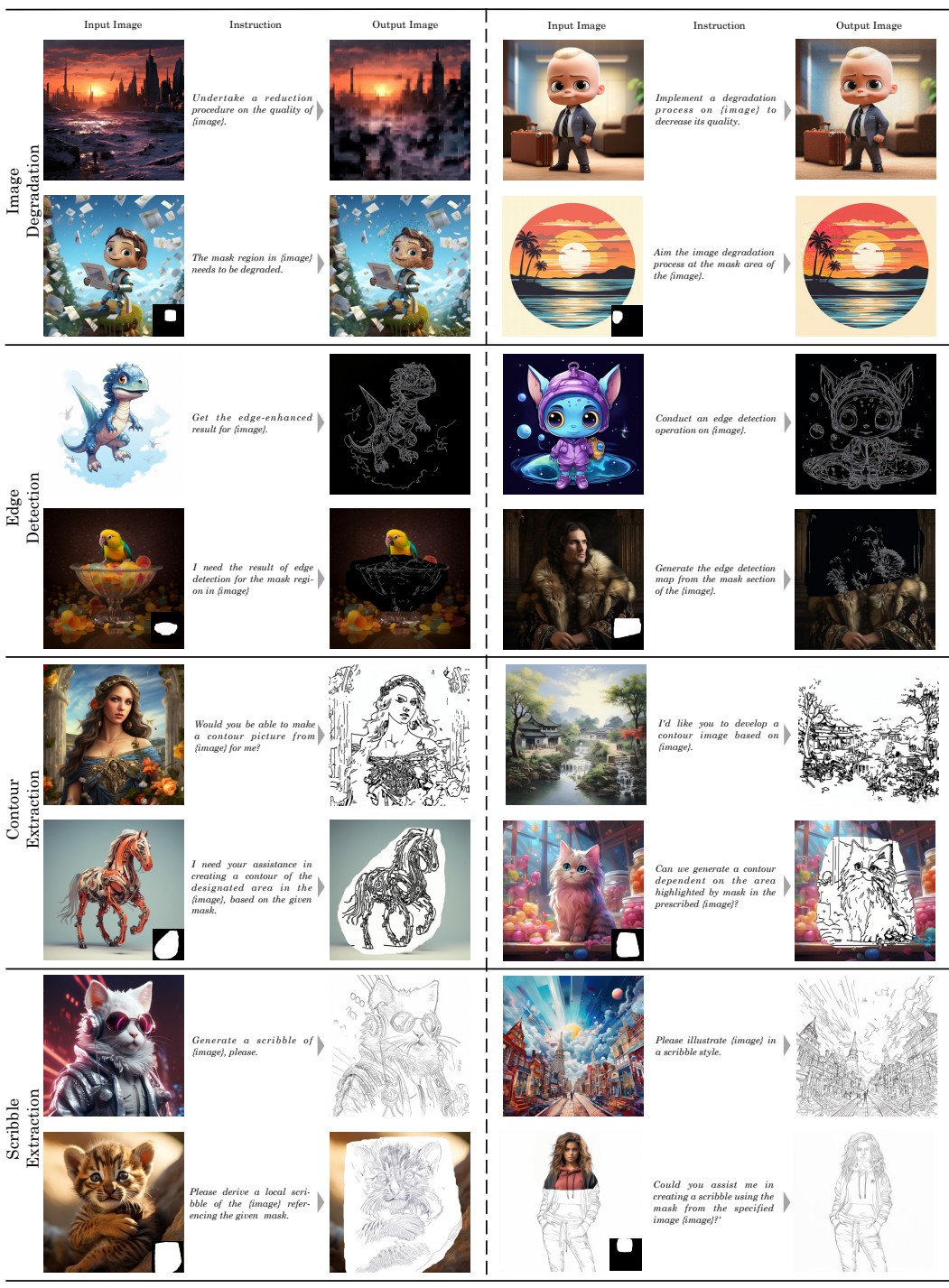

Figure 37: The ACE's generated visualization of image degradation, edge extraction, contour extraction, and scribble extraction in low-level visual analysis.

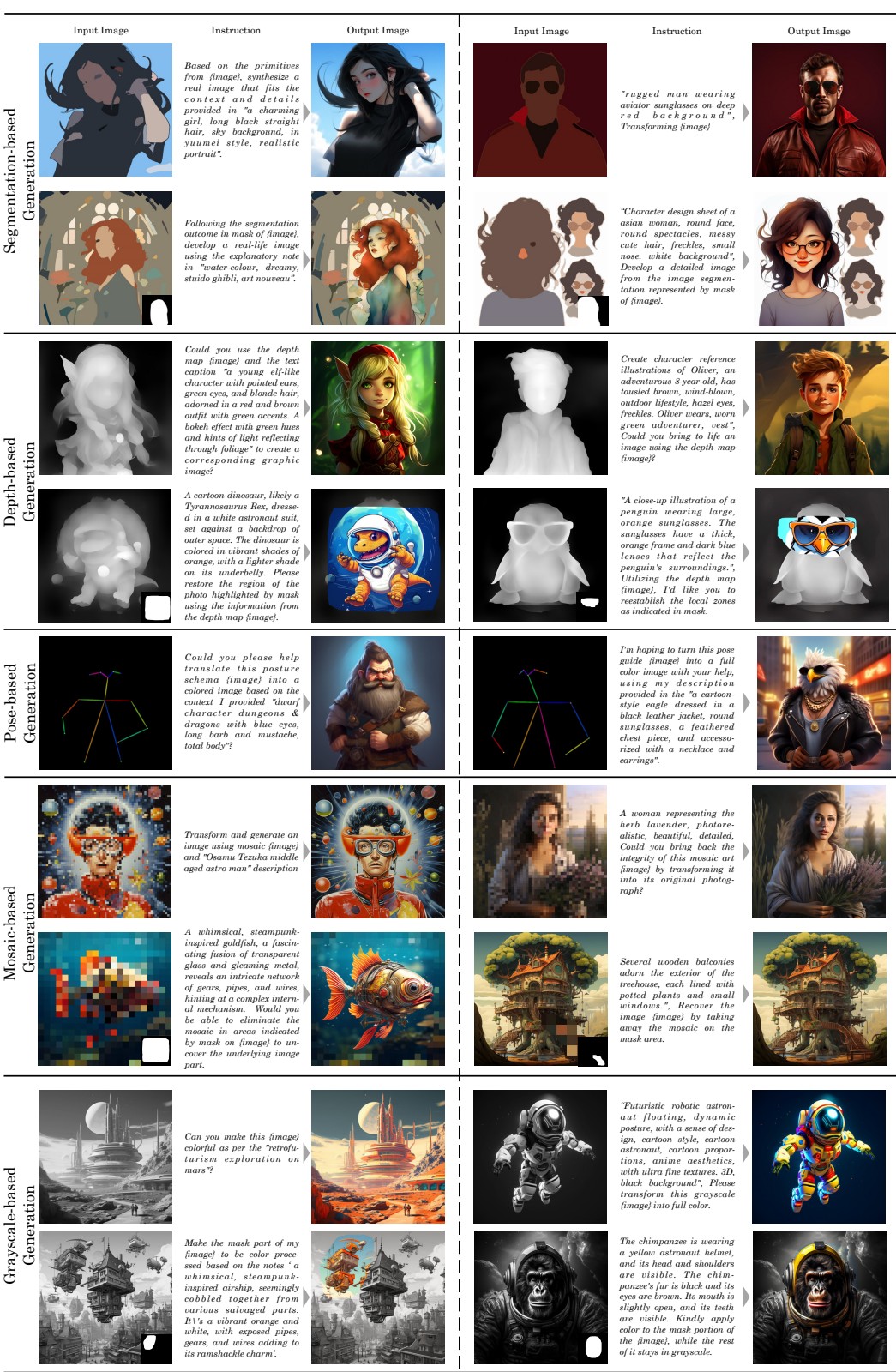

Figure 38: The ACE's generated visualization of segmentation-based, depth-based, pose-based, mosaic-based, and grayscale-based generation in controllable generation.

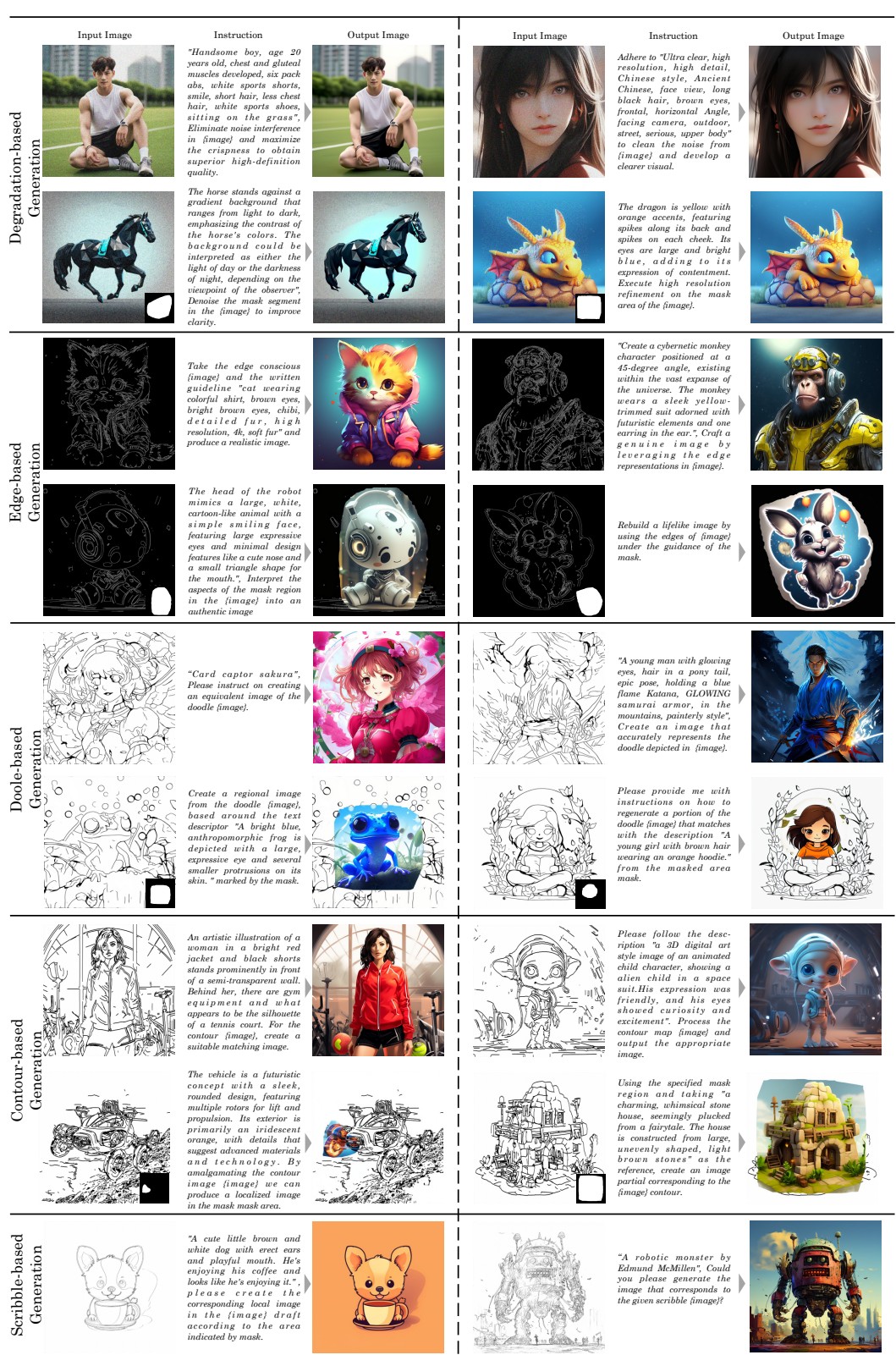

Figure 39: The ACE's generated visualization of degradation-based, edge-based, doodle-based, contour-based, and scribble-based generation in controllable generation.

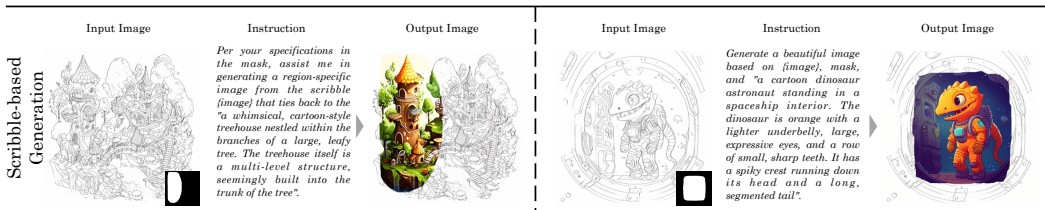

Figure 40: The ACE's generated visualization in scribble-based controllable generation.

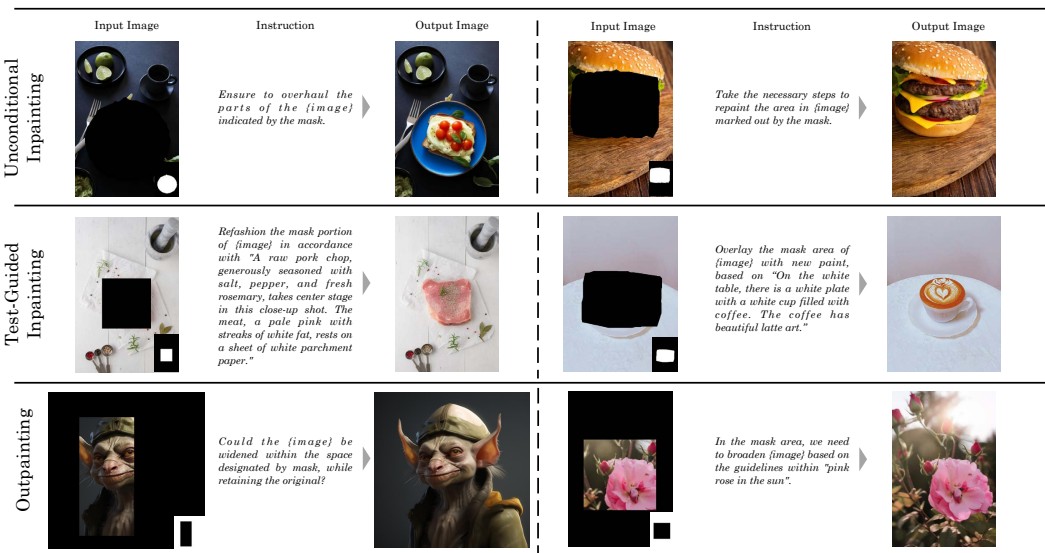

Figure 41: The ACE's generated visualization of repainting.

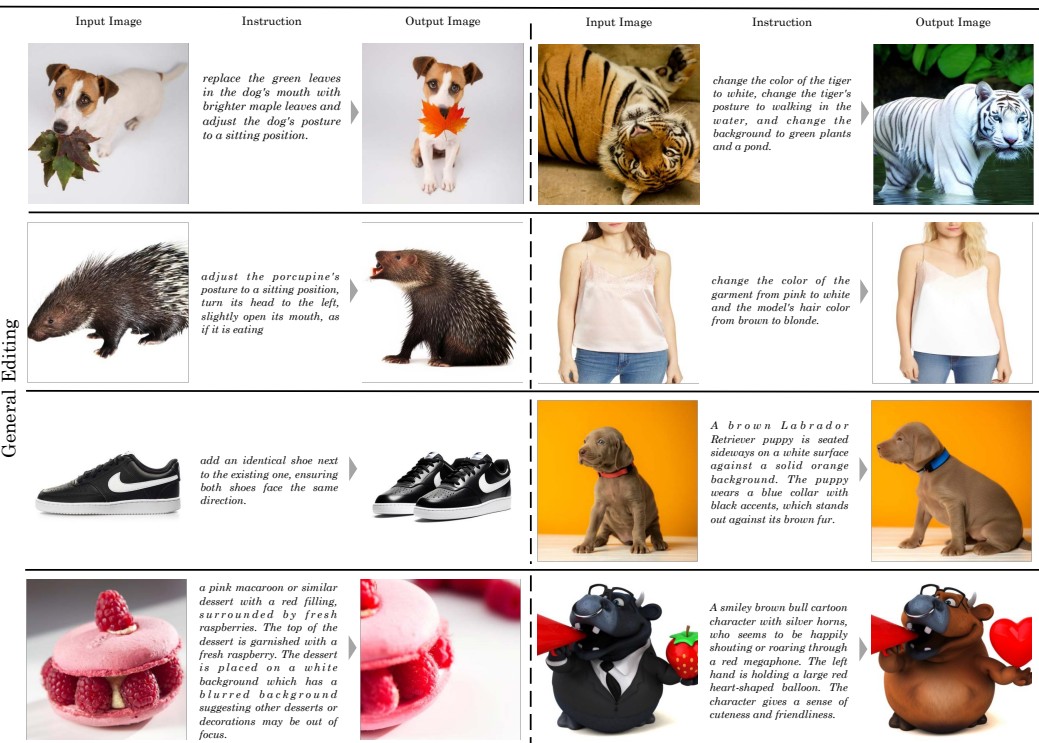

Figure 42: The ACE's generated visualization of general editing in semantic editing.

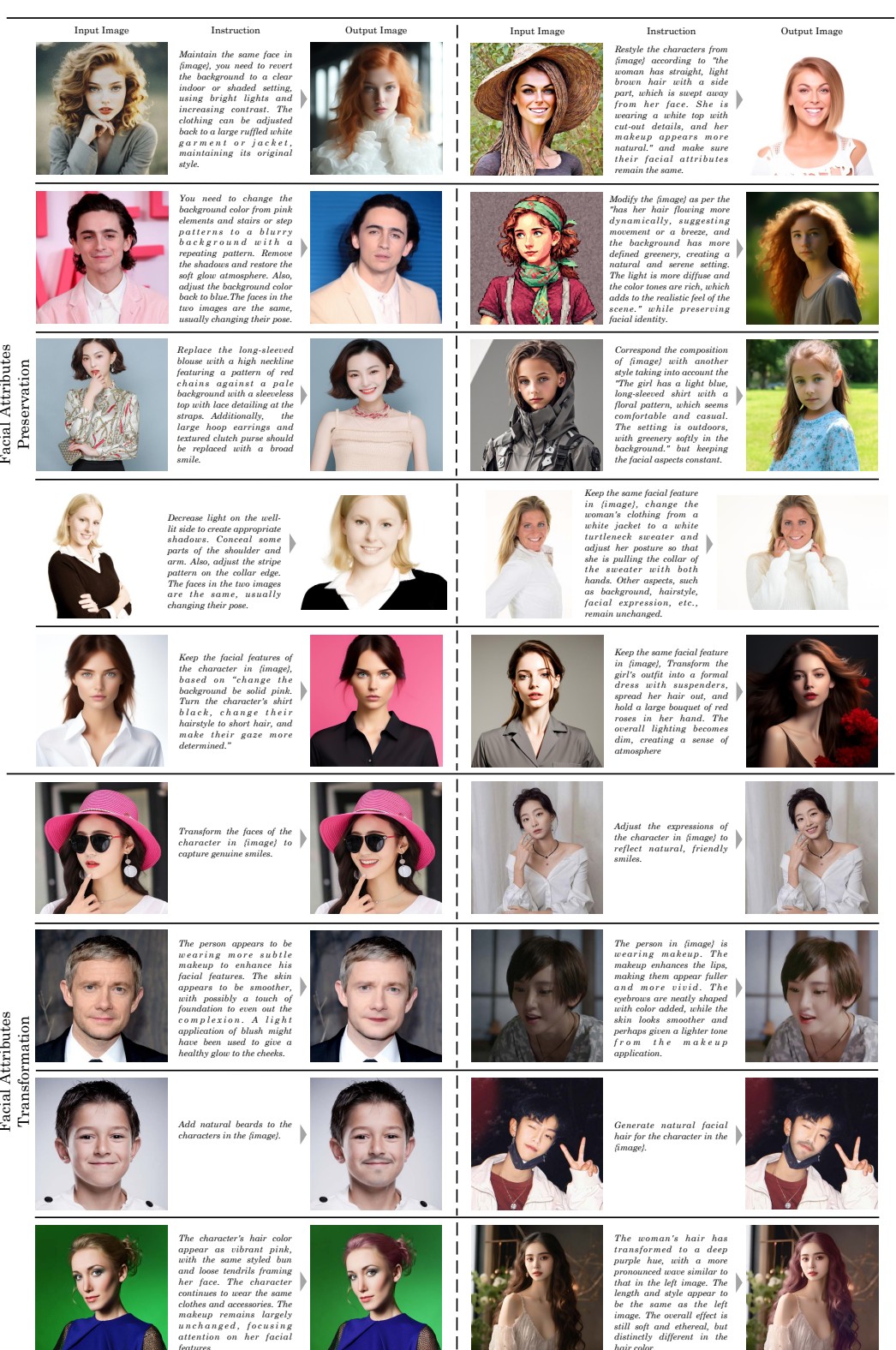

Figure 43: The ACE's generated visualization of facial editing in semantic editing.

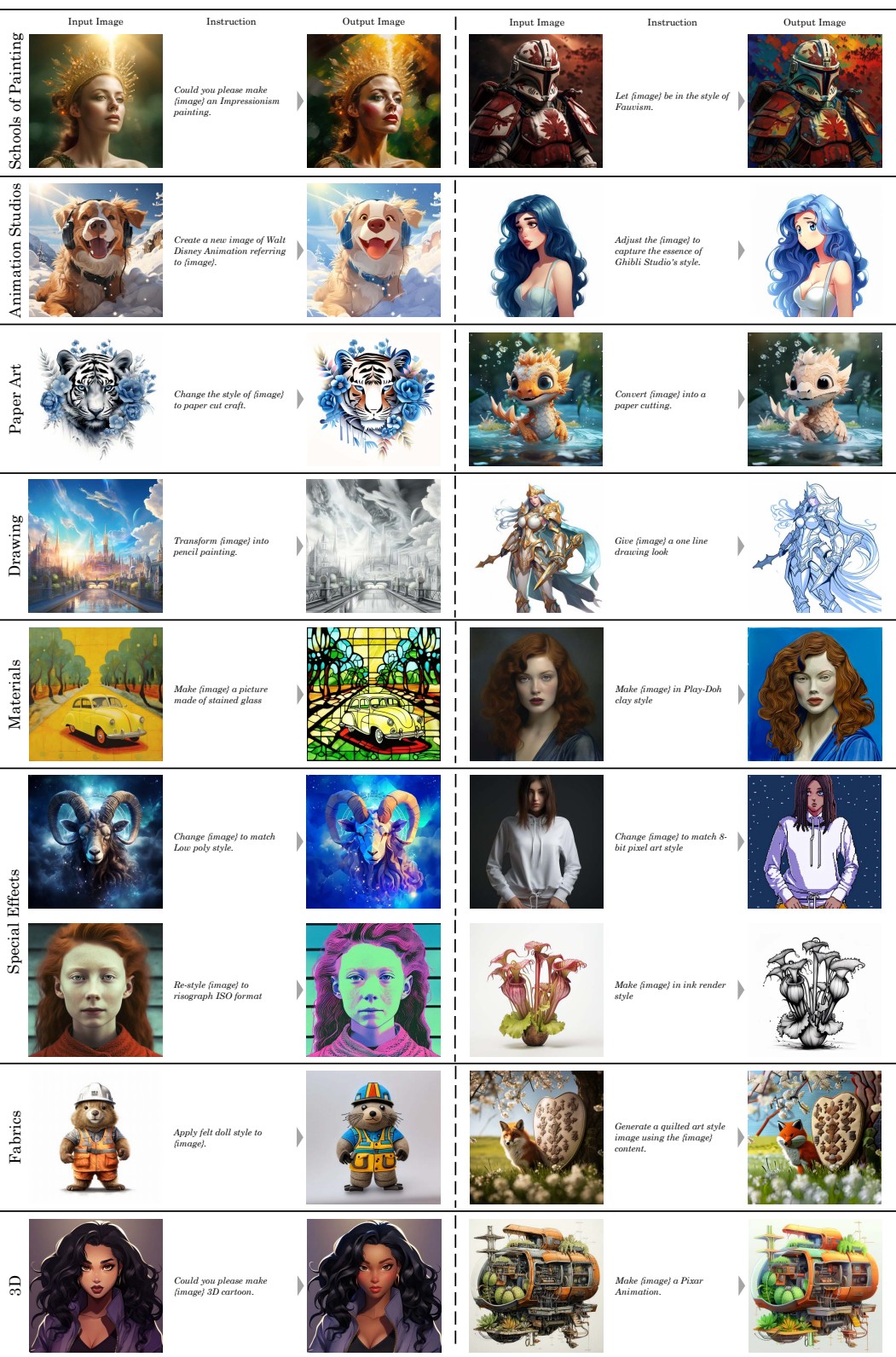

Figure 44: The ACE's generated visualization of style editing in semantic editing.

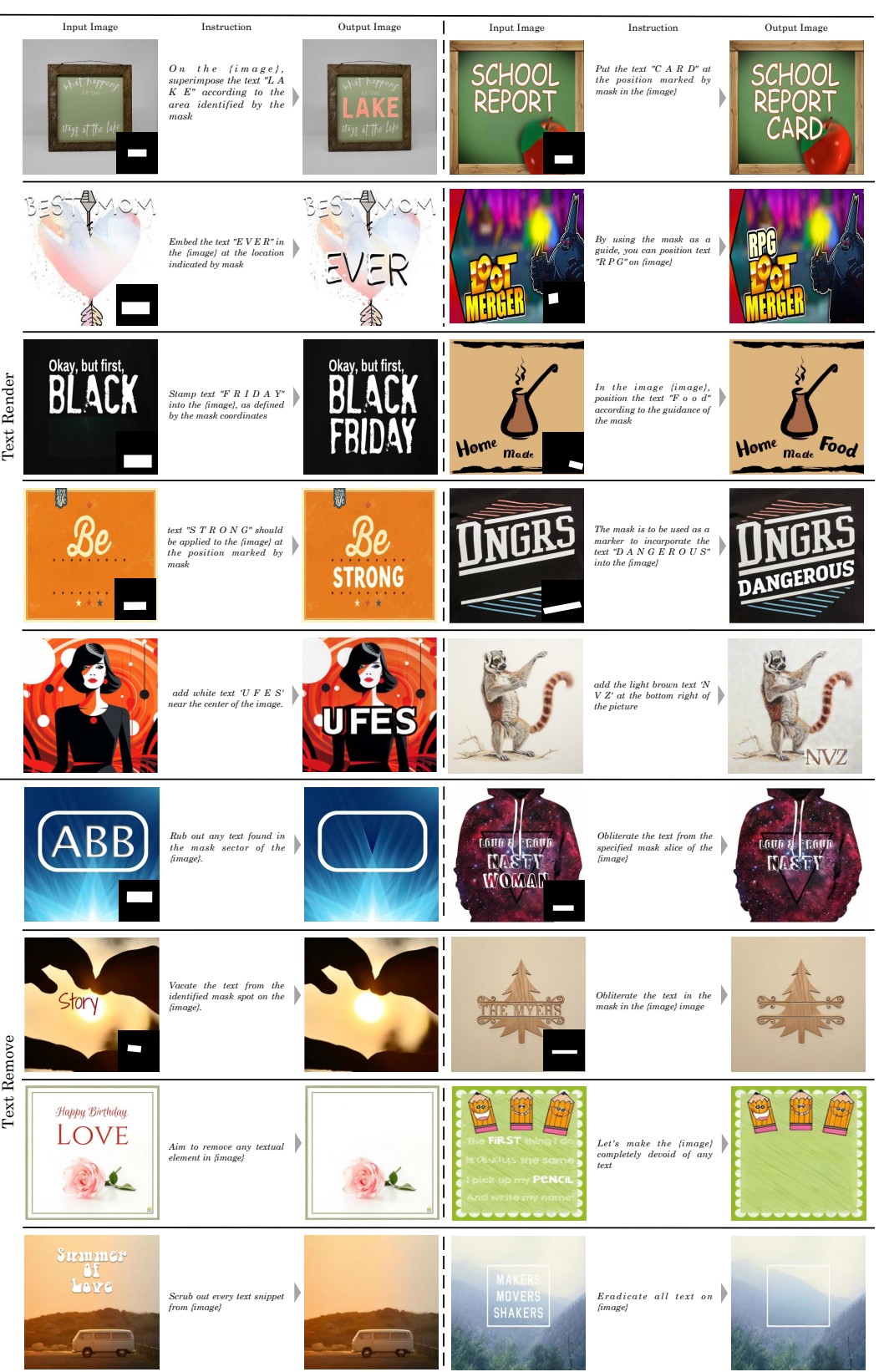

Figure 45: The ACE's generated visualization of text editing in element editing.

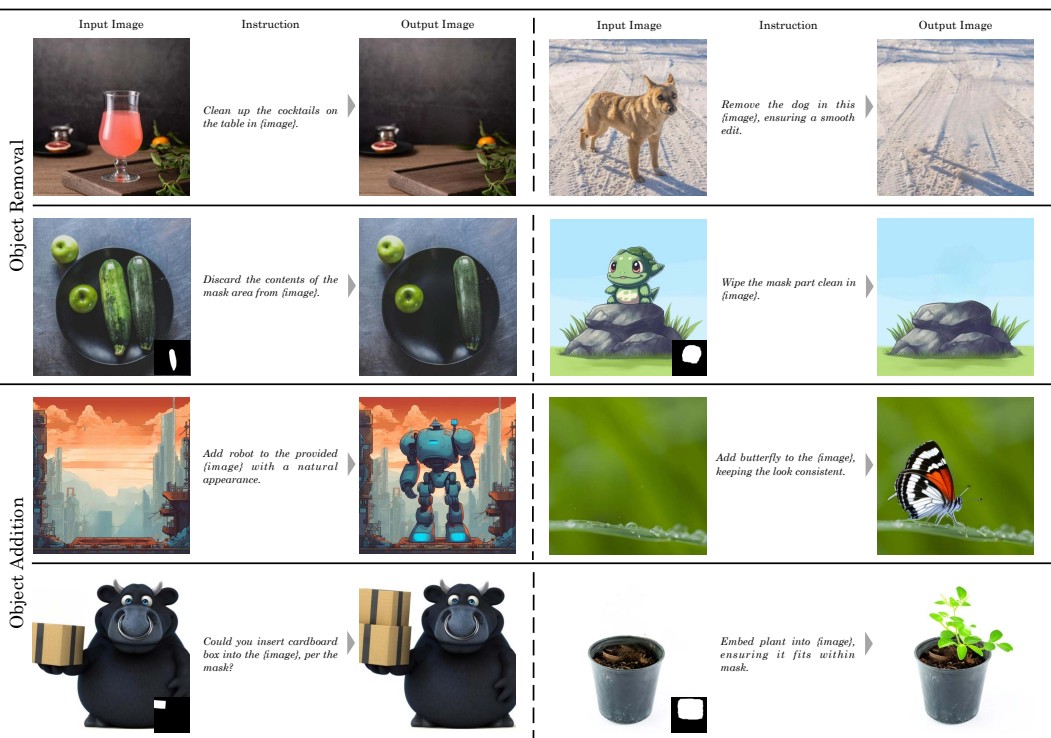

Figure 46: The ACE's generated visualization of object editing in element editing.

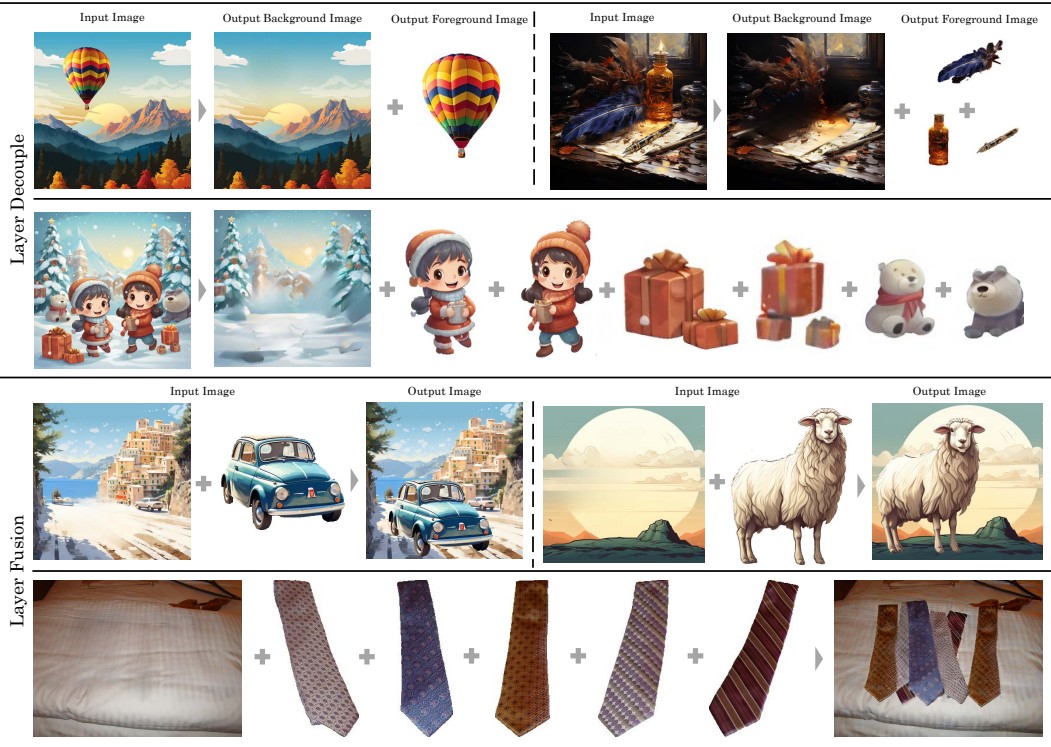

Figure 47: The ACE's generated visualization of layer decouple and layer fusion in layer editing.

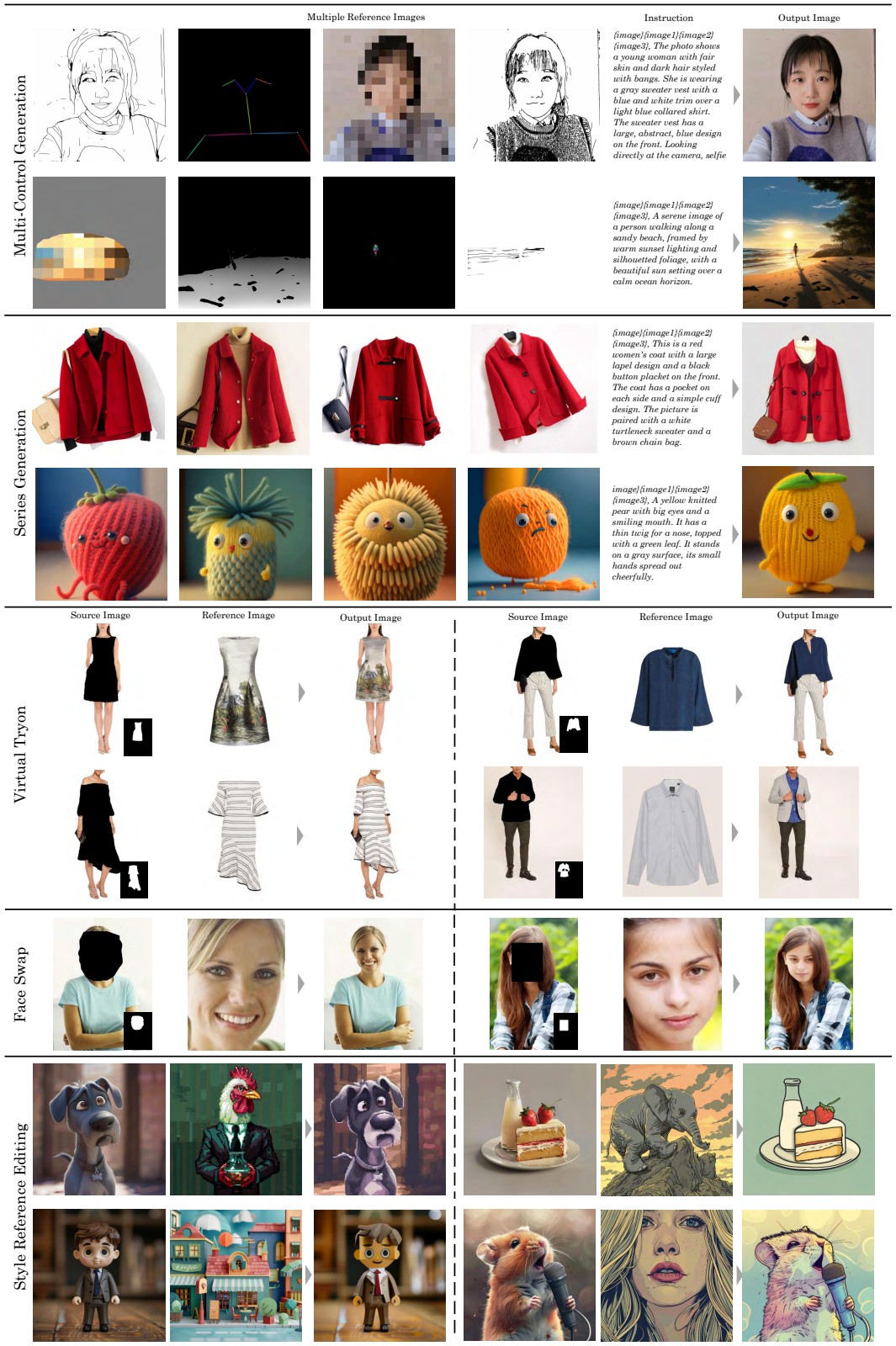

Figure 48: The ACE's generated visualization of multi-reference generation and reference-guided editing.

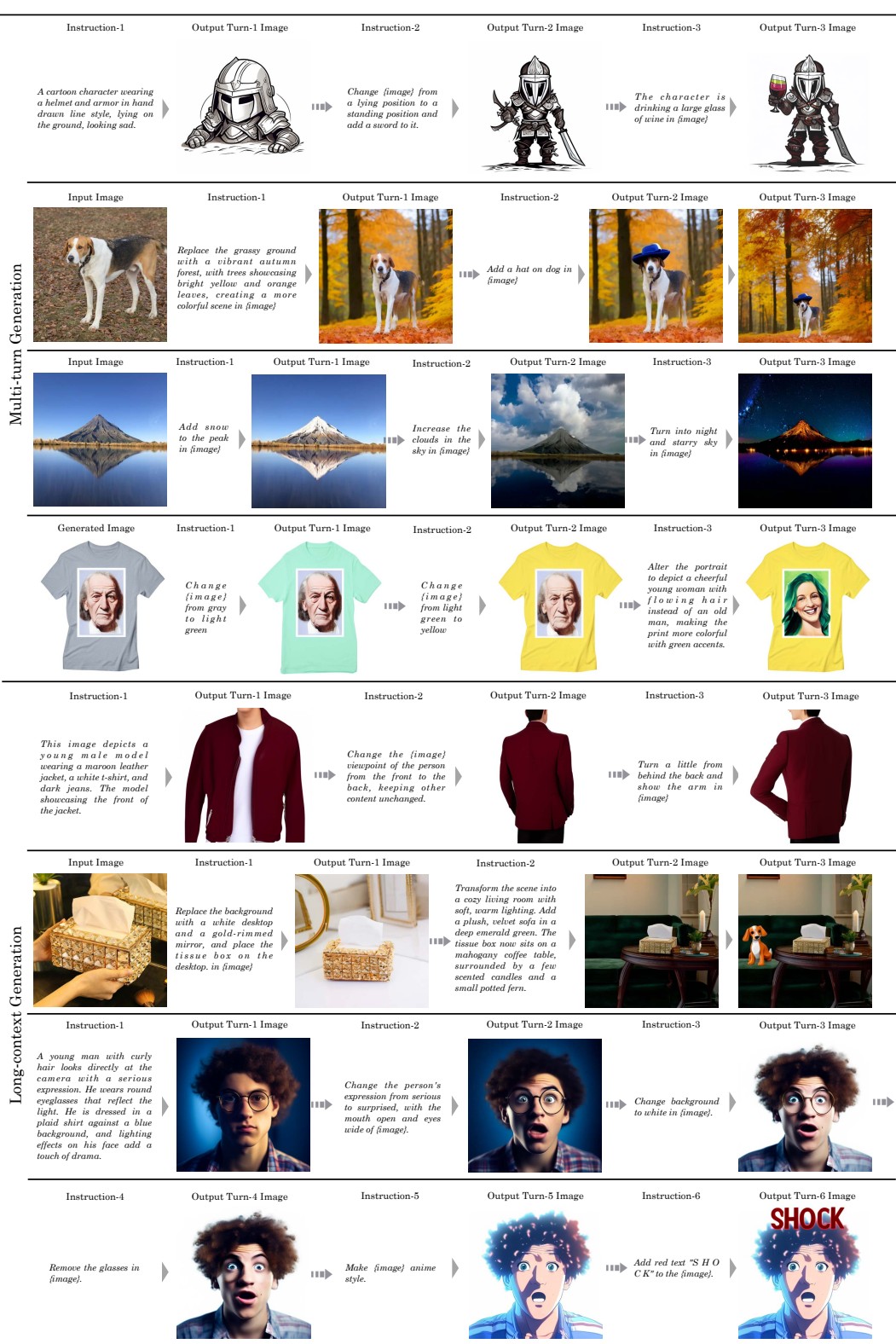

Figure 49: The ACE's generated visualization of multi-turn and long-context generation.