# OpenReview forum: "ACE: All-round Creator and Editor Following Instructions via Diffusion Transformer"
_ICLR.cc/2025/Conference — ICLR 2025 Poster_

### Official Review · Reviewer_7X1X · 2024-10-31

**Soundness:** 3
**Presentation:** 4
**Contribution:** 3
**Rating:** 6
**Confidence:** 4

**Summary:**

This paper introduces ACE (All-round Creator and Editor), a unified foundational model capable of handling a diverse array of visual generation tasks. By incorporating Long Contextual Units (LCU) and an efficient multimodal data collection methodology, ACE demonstrates exceptional performance in multi-task joint training, encompassing a wide range of tasks from text-guided generation to iterative image editing. Experimental results indicate that ACE significantly outperforms existing methods across multiple benchmark tests, showcasing its robust potential for practical applications.

**Strengths:**

ACE introduces LCU, a novel approach that unifies various modal conditions, enabling the model to handle complex multimodal tasks. LCU allows ACE to flexibly adapt to different tasks, including generation and editing, which is lacking in current models. By integrating historical information into LCU, ACE can handle multi-turn editing tasks, enhancing its practicality in continuous interaction scenarios. ACE covers eight basic generation tasks and supports multi-turn and long-context tasks, establishing a comprehensive evaluation benchmark, significantly outperforming existing methods, especially in image editing tasks. User studies show that ACE is more in line with human perception. This paper not only makes significant contributions and proposes a practical and innovative solution but also excels in writing and figure drawing, with clear diagrams and rigorous logic, providing an excellent reading experience for the audience.

**Weaknesses:**

Model Efficiency and Scalability:
The paper should include a more detailed discussion on the computational efficiency and scalability of the model:
It is important to evaluate the model's performance when processing large-scale data to understand its practical applicability.

In-depth Analysis of Specific Tasks:
The paper should provide a thorough performance analysis for specific tasks.
This analysis should include comparisons with models that are specifically designed for those tasks, such as image editing models and inpainting models.

Data Annotation Quality:
While MLLM-assisted annotation improves efficiency, the quality of automatic annotations may not always be on par with manual annotations.
A quantitative analysis of the data annotation quality would enhance the credibility of the paper.

**Questions:**

Discussion on Model Efficiency and Scalability: Could you provide more details on the model's performance across different scales of data? This would help in understanding its computational efficiency and scalability.

In-depth Analysis of Specific Tasks: For key tasks, could you offer a detailed comparison with state-of-the-art models specifically designed for those tasks? This would provide a clearer picture of the model's relative performance.

Enhancing Model Interpretability: Could you explore the decision-making process of the model and provide an interpretability analysis of the generated results? This would help in understanding how the model arrives at its outputs.

---

> ### Author Response · Authors · 2024-11-21
>
> Dear Reviewer 7X1X,
>
> Thank you for acknowledging the contributions and presentations. We address your concerns as follows:
>
> ---
>
> **Q1: Discussion on Model Efficiency and Scalability**
>
> We incorporate more information about computational efficiency during training and inference in the *"Supplementary Material, Section IMPLEMENTATION DETAILS"*.
>
> In general, the training and inference efficiency is related to visual sequence length and input image number (*"SubSection Computational Efficiency"*). This is the reason why we training the model with a multi-stage training strategy, which is training with a small visual sequence length and fewer input images at first and increasing the length and the number of input images in the following stages. (see in Fig.28-a)
>
> We further conduct evaluations of our intermediate model checkpoints on the MagicBrush benchmark to evaluate the impact of data scale in the *"SubSection Checkpoints Evaluation"*. Generally, the model’s performance improves when trained with more data. (see in Fig.28-b)
>
> ---
>
> **Q2: In-depth Analysis of Specific Tasks**
>
> We conduct detailed quantitative comparisons with specifically designed state-of-the-art methods for key tasks: facial editing and local text rendering, please refer to *"Supplementary Material, Section MORE EXPERIMENTS, SubSection Facial Editing and Local Text Render"* for the details. More qualitative comparisons for inpainting and controllable generation are added to *"Supplementary Material, Section MORE EXPERIMENTS, SubSection More Qualitative Comparison"*.
>
> ---
>
> **Q3: Data Annotation Quality**
>
> As shown in Fig. 4, we used Qwen-VL for the initial construction of instructions and obtained more accurate instruction descriptions through manual annotations. We then utilized this high-quality data for training the InternVL model. It is worth noting that the entire process is conducted in an iterative update manner, allowing us to continually refine our annotated data and iteratively enhance the performance of the fine-tuned InternVL. Furthermore, based on our sampling evaluation, the accuracy of the model's instruction annotations has reached over 92 %, which is sufficient for use as training pair data. For further improvement in annotation accuracy, it is necessary to address the deficiencies in detailed image descriptions within the multimodal model itself, which is also a challenge faced by current pure image labeling models.
>
> ---
>
> **Q4: Enhancing Model Interpretability**
>
> In *"Supplementary Material, Section ARCHITECTURE DESIGN"*, we outline our key architectural design: Long-context Attention Block and Image Indicator Embeddings, as well as a qualitative analysis of these components. This may partially explain the model's functionality. We also visualize the editing process by decoding intermediate model outputs during the de-noising process and try to explain the model's behavior at each step in *"Supplementary Material, Section IMPLEMENTATION DETAILS, SubSection Visualization of Editing Process."*. When we use the instruction “Let a car appear in {image}.” to edit the image, the model identifies the area to be edited in the initial steps and subsequently copies the unchanged regions from the input image in the following steps. In the steps leading up to the final stage, additional details are incrementally added to the edited area until completion.
>
> ---

---

> ### Author Response · Authors · 2024-12-03
>
> Dear Reviewer 7X1X,
>
> We hope this message finds you well. We sincerely appreciate the time and effort you have dedicated to reviewing our submission.
>
> We have submitted our rebuttal and would like to follow up to inquire whether our responses have sufficiently addressed your concerns. Please let us know if you have any remaining questions or require additional clarification.
>
> Best regards,
>
> Authors of Submission 8711

---

### Official Review · Reviewer_bd3V · 2024-11-02

**Soundness:** 3
**Presentation:** 3
**Contribution:** 3
**Rating:** 6
**Confidence:** 4

**Summary:**

This work proposes a unified visual generation and editing framework that supports a wide range of predefined tasks. To train and evaluate the proposed ACE, this work also introduces a data curation pipeline and an overall benchmark. Experimental results and numerous use cases demonstrate the superiority of the proposed method.

**Strengths:**

The method provides a unified visual generation and editing framework that supports a wide range of predefined tasks.
 The benchmark is comprehensive, designed to evaluate visual generation and editing models effectively.

**Weaknesses:**

The paper lacks some ablation studies to help readers understand the authors' design choices. Additionally, the results in Table 2 may not be entirely fair, as the superiority of ACE might be attributed to the scale of the data.

**Questions:**

1. What would happen if the Text Encoder T5 were replaced with an LLM? Would it be able to understand more diverse instructions?
2. Will the collected data be made public?

---

> ### Author Response · Authors · 2024-11-21
>
> Dear Reviewer bd3V,
>
> Thank you for acknowledging the proposed method and experiments. We address your concerns as follows:
>
> ---
>
> **Q1: Architectural Design**
>
> Thank you for your suggestions. We have outlined the key architectural design in the *"Supplementary Material, Section ARCHITECTURE DESIGN"*.
>
> Our model is anchored by two principal components: the Long-context Attention Block and Image Indicator Embeddings. The Long-context Attention Block is specifically designed to handle input image sequences of varying lengths, addressing the limitations of the conventional attention block employed in DiT, which cannot effectively manage sequences of disparate lengths. Meanwhile, the Image Indicator Embeddings facilitate the alignment between images in the sequence and their corresponding mentions within the text prompt. A qualitative analysis of these components is also provided in the ARCHITECTURE DESIGN Section.
>
> ---
>
> **Q2: Results in Tab. 2**
>
> Unlike previous methods that primarily focus on limited editing tasks, our approach is designed as a foundational editing model capable of addressing a broad spectrum of editing tasks. Consequently, leveraging large-scale data is a logical choice to enhance our model's performance. Moreover, most existing edit models are derived from foundational text-to-image generation frameworks, which have been trained on extensive data. Therefore, conducting a fair comparison under equivalent training data conditions is not feasible.
>
> ---
>
> **Q3: Replacement of Text Encoders**
>
> In the *"Supplementary Material, Section DISCUSSION, SubSection Future Work"*, we also mentioned that introducing LLMs or MLLMs could potentially help us better understand user intentions within general instructions.
>
> Regarding the design of text encoders, there are currently two design approaches. One is based on diffusion models specifically designed for text-to-image tasks, such as the MagicBrush and CosXL methods that utilize models from the SD series; these models primarily rely on CLIP or T5 for text encoding. The other approach is based on multimodal models, such as Llama and Phi, which are pre-trained to acquire generative and editing capabilities, like Seed-X, Seed-X Edit methods. The former naturally has the capacity to generate high-quality images, while the latter exhibits superior semantic understanding. Our future exploration will focus on how to combine and enhance the strengths of both approaches.
>
> ---
>
> **Q4: Public Content**
>
> We will make the model, training code, inference code, chatbot, and evaluation benchmark publicly available. However, due to organizational policies, we are unable to disclose the training data.
>
> ---

---

> ### Author Response · Authors · 2024-12-03
>
> Dear Reviewer bd3V,
>
> We hope this message finds you well. We sincerely appreciate the time and effort you have dedicated to reviewing our submission.
>
> We have submitted our rebuttal and would like to follow up to inquire whether our responses have sufficiently addressed your concerns. Please let us know if you have any remaining questions or require additional clarification.
>
> Best regards,
>
> Authors of Submission 8711

---

### Official Review · Reviewer_vndP · 2024-11-04

**Soundness:** 3
**Presentation:** 2
**Contribution:** 3
**Rating:** 6
**Confidence:** 3

**Summary:**

1. Propose ACE, a unified foundational model framework that supports a wide range of visualgeneration tasks, achieve a best task coverage.
2. Define the CU for unifying multi-modal inputs across different tasks and incorporate long context CU.
3. Design specific data construction pipelines for various tasks to enhance the quality and eff-ciency of data collection.
4. Establish a more comprehensive evaluation benchmark compared to previous ones, cover-ing the most known visual generation tasks.

It's a lot of work, from method to data to data construction pipelines to benchmark, very systematic and complete work.
And all-in-one models are really interesting and is consistent with the general trend of generate model development.
But,
drawings are terrible, and the method is a little weak. Maybe you could use the all-in-one methods in low-level works as reference.

**Strengths:**

1. Propose ACE, a unified foundational model framework that supports a wide range of visualgeneration tasks, achieve a best task coverage.
2. Define the CU for unifying multi-modal inputs across different tasks and incorporate long context CU.
3. Design specific data construction pipelines for various tasks to enhance the quality and eff-ciency of data collection.
4. Establish a more comprehensive evaluation benchmark compared to previous ones, cover-ing the most known visual generation tasks
5.Analyze and categorize these conditions from textual and visual modalities respectively, includeTextual modality and Visual modality.

**Weaknesses:**

1. The drawings are terrible!!! In particular, Figure 3. Incongruous text proportions and strange colour scheme...It's in the lower-middle range of T2I work.
2. The method is a little weak. All-in-one methods have been far dicussed in the field of low-level and it's ripe for the picking. Compared with them, the ACE module is not that impressive.

**Questions:**

Please DRAW better.
I don't find the computing resource? I think it would be big, maybe you could have a discuss.
Related work? i think there should be other works that make try building an all-in-one visual generation model. Maybe you could list them clearly, I'm not an expert on this.

---

> ### Author Response · Authors · 2024-11-21
>
> Dear Reviewer vndP,
>
> Thank you for your time and suggestions. We address your concern as follows:
>
> ---
>
> **Q1: About Drawing**
>
> Thank you for your feedback. We hope that the chosen text and colors can help readers better understand the content without causing any misunderstandings. Throughout the paper, we have adopted a consistent font and specific shades (blue, yellow, and green) as the baseline pattern, while using lighter and darker corresponding shades as accents to ensure visual harmony and aesthetics. All figures adhere to this principle as much as possible.
>
> Additionally, Reviewer vGPZ and bd3V believe that our presentation is *good*, while Reviewer 7X1X considers it *excellent* and further notes that "*it excels in writing and figure drawing, with clear diagrams and rigorous logic, providing an excellent reading experience for the audience*". We will continue to strive to improve our visual presentation and appreciate your understanding.
>
> ---
>
> **Q2: ACE vs. Other All-in-One**
>
> ACE is an All-round **Creator** and **Editor**, that supports a wide range of visual generation tasks, including 8 basic types: Text-guided Generation, Low-level Visual Analysis, Controllable Generation, Semantic Editing, Element Editing, Repainting, Layer Editing, and Reference Generation.
>
> The low-level tasks you mentioned are only **a very small part** of what we focus on. Furthermore, the Low-level Visual Analysis described in the manuscript includes *Image Segmentation, Depth Estimation, Human-pose Estimation, Image Mosaic, Image Degradation/Super-Resolution, Image Grayscale, Edge Detection, Doodle Extraction, Contour Extraction, and Scribble Extraction*. I have not been able to find any existing work that uses a single model to handle all of these tasks.
>
> In addition to handling low-level tasks, we also have many other applications, such as basic text-to-image generation, comprehensive controllable generation, instruction-based editing, reference-guided generation, and long-context-guided generation. To support all of these functions, an all-in-one approach needs to be redesigned and cannot be directly referenced from the low-level domain you mentioned.
>
> There were some all-in-one controllable generation methods, such as Uni-Controlnet[1] and Controlnet-Union-SDXL[2]. However, these methods merely provide a controllable generation all-in-one model from a unified perspective, which only accounts for a part of our tasks. Furthermore, we also compared several general editing methods, as detailed in Fig. 5, including IP2P[3], MagicBrush[4], CosXL[5], SEED-X[6], and UltraEdit[7].
>
> As you mentioned, this is a lot of work, as we need to manage the data for various tasks separately, especially since acquiring data for the more advanced tasks is much more challenging compared to low-level tasks (*"Supplementary Material, Section DATASETS DETAIL*"). At the same time, we have to design a unified method that covers these tasks.
>
> We appreciate your feedback and look forward to it.
>
> [1] Zhao et al. "Uni-ControlNet: All-in-One Control to Text-to-Image Diffusion Models." NeurIPS 2023.
>
> [2] xinsir, et al. "controlnet-union-sdxl-1.0." Hugging Face.
>
> [3] Brooks et al. "InstructPix2Pix: Learning To Follow Image Editing Instructions." CVPR2023.
>
> [4] Zhang et al. "MagicBrush: A Manually Annotated Dataset for Instruction-Guided Image Editing."
> NeurIPS 2023.
>
> [5] StabilityAI. "CosXL." Hugging Face.
>
> [6] Ge et al., "SEED-Data-Edit Technical Report: A Hybrid Dataset for Instructional Image Editing." arXiv 2024.
>
> [7] Zhao et al. "UltraEdit: Instruction-based Fine-Grained Image Editing at Scale." arXiv 2024.
>
> ---
>
> **Q3: Further Information**
>
> For more information, please refer to the *"Supplementary Material"*. In this material, we have dedicated a significant amount of space to provide a more detailed description of the methods and to showcase additional visual results.
>
> Regarding the computing resources you mentioned, we used A800 as the base hardware and dynamically adjusted the quantity used according to the training to meet the expected batch size. For further details, please refer to *"Supplementary Material, Section IMPLEMENTATION DETAILS"*. Additionally, related work can also be found in *"Supplementary Material, Section RELATED WORK"*.
>
> ---

---

> ### Author Response · Authors · 2024-12-03
>
> Dear Reviewer vndP,
>
> We hope this message finds you well. We sincerely appreciate the time and effort you have dedicated to reviewing our submission.
>
> We have submitted our rebuttal and would like to follow up to inquire whether our responses have sufficiently addressed your concerns. Please let us know if you have any remaining questions or require additional clarification.
>
> Best regards,
>
> Authors of Submission 8711

---

### Official Review · Reviewer_vGPZ · 2024-11-04

**Soundness:** 3
**Presentation:** 3
**Contribution:** 4
**Rating:** 8
**Confidence:** 4

**Summary:**

This work proposes an All-round Creator and Editor as a unified foundation model for visual generation tasks. The main technical contribution lies in introducing a Long-context Condition Unit that standardizes diverse input formats. Built upon diffusion transformers, the architecture incorporates condition tokenizing, image indicator embedding, and long-context attention blocks to achieve unified visual generation capabilities. To address the scarcity of training data, the authors develop a data collection pipeline that combines synthesis/clustering-based approaches. Additionally, they establish a comprehensive benchmark for evaluating model performance across various visual generation tasks.

**Strengths:**

1) The framework unifies multiple image generation and editing tasks through a single model, avoiding the hassle of calling separate specialized models. The proposed LCU provides a structured approach to incorporating historical context in visual generation.

2) The paper presents systematic methodologies for data collection and instruction construction, which contributes to the development of all-in-one visual generative foundation models.

3) The evaluation benchmark provides comprehensive coverage across diverse image manipulation and generation tasks, enabling thorough performance assessment.

**Weaknesses:**

Technical Issues:
1. Formatting inconsistencies: in lines 417-418, the image placement obscures instruction text.

2. The authors are encouraged to provide discussions on task-specific performance trade-offs during training, specifically how optimizing for one task might affect the performance of others.

3. It would be helpful to provide methodological details regarding parameters in data preparation (lines 321-325), such as cluster number determination and data cleaning criteria.

4. The qualitative results in Figure 5 reveal some limitations. 1) Row 1 (left): ACE generates a distorted hand. 2)  Row 2 (right) and Row 4 (left): The model exhibits undesired attribute modifications not specified in the instructions, including unintended gesture alterations / head rotation changes, and camera perspective shifts.

**Questions:**

1. Regarding Figure 6, the authors are encouraged to elaborate on the empirical or theoretical basis for the chosen data distribution and its specific advantages for the ACE model.

2. The paper would benefit from addressing the practical challenges of model updates. Specifically, how might one efficiently incorporate new functionalities without complete model retraining? This consideration is crucial for the model's practical deployment and ongoing development.

**Details Of Ethics Concerns:**

The paper mentions that the internal dataset was used for training, which may involve issues related to portraits and copyrighted images.

---

> ### Author Response · Authors · 2024-11-21
>
> Dear Reviewer vGPZ,
>
> Thank you for acknowledging our contributions and your valuable comments. We address your concern as follows:
>
> ---
>
> **Q1: Formatting**
>
> Thanks. We have completed the corrections in the manuscript.
>
> ---
>
> **Q2: Task Interdependence**
>
> Thank you for your suggestions. We have added a discussion on task interdependence in the *"Supplementary Materials, Section IMPLEMENTATION DETAILS, SubSection Task Interdependence"*. The main details are as follows:
>
> In an all-in-one visual generation model, there exist multiple interactions between tasks, similar to that in large language models, and this relationship can be viewed as a complex balancing action.
>
> i) Complementarity between tasks: The combined influence of various tasks can lead to a certain degree of generalized behavior across tasks. For instance, in the style transfer task, our prepared data and training process focus on pixel-aligned global image transfer. However, by incorporating learnings from other tasks related to mask guidance or subject guidance, the model can acquire the ability to perform style transfer in localized areas. (as in Fig. 29)
>
> ii) Competition between tasks: As the scale of tasks increases, the potential for competition also grows, particularly in scenarios where user instructions are ambiguous. For example, when adding the text "APPLE" to an image, it is essential to specify that it is text to be added; otherwise, due to semantic ambiguity, the result may instead involve the addition of an object depicting an apple. (as in Fig. 29)
>
> To achieve optimal performance balance, we first focus on adjusting the data sampling rates for each task in a phased manner during the training process, monitoring this through a validation set. Additionally, more detailed descriptions of instructions are needed in the preparation of training data to prevent semantic confusion between tasks. Through these methods, we aim to ensure that the model can fully leverage the complementarity between different tasks while controlling for any potential negative impacts.
>
> However, the relationships between different tasks still require further exploration to better optimize the model's performance. Future work will also focus on how to effectively evaluate and adjust these influencing factors to achieve a more balanced and comprehensive execution of tasks.
>
> ---
>
> **Q3: Details of Data Processing**
>
> We provide a detailed analysis and parameter description regarding this issue in the *"Supplementary Materials, Section IMPLEMENTATION DETAILS, SubSection Data Preprocessing Details."*.
>
> In the data preparation stage, our main considerations for parameter selection are computational efficiency and ensuring high data relevance. The designed hierarchical aggregation pipeline for pairing content-related images involves clustering, identifying first-level disjoint sets, and determining second-level disjoint sets.
>
> Initially, data is clustered into 10, 000 clusters using K-means clustering based on SigLip features, allowing the Union-Find algorithm to be executed more efficiently by keeping the data scale under 100K one node in the parallel execution. First-level disjoint sets are formed by analyzing the similarities of SigLip features within these clusters, using a SigLip similarity matrix and thresholds for data pruning to ensure strong internal connections. Second-level disjoint sets are established through task-specific correlations, with specialized models used for various tasks such as background alterations or ID preservation, applying different similarity measures and thresholds to maintain the necessary correlation. This process utilizes advanced data mining and correlation models tailored to specific tasks, employing techniques like binary classification with the ViT-B-16-SigLIP and cosine distance for facial features.
>
> ---

---

> > ### Author Response · Authors · 2024-11-21
> >
> > ---
> >
> > **Q4: Results in Fig.5**
> >
> > i）As stated in the Limitations subsection of the *"Supplementary Materials, Section DISCUSSION"*, the quality of our model's generation is constrained by its scale (only 0.6B), which may lead to some common issues often encountered in image generation tasks. Increasing the model's scale can effectively alleviate this issue.
> >
> > ii）ACE can intentionally control whether the generation is aligned or unaligned through instructions. For example, methods like InstanceID and CosXL are aligned, keeping the character's pose mostly unchanged while altering their style or background. In contrast, IP-Adapter and FaceChain retain the core facial features, with other content being controlled by the text. By using descriptive instructions, we can achieve more precise control over the generated results or present a diverse array of content.
> >
> > ---
> >
> > **Q5: Training Data Distribution**
> >
> > The training data used for ACE depends on the following aspects:
> >
> > ii）Ease of Data Acquisition: Tasks such as low-level Visual Analysis and Repainting rely on on-the-fly processing flows that can be easily obtained, while conditional generation data can be derived from various conditional models. In contrast, tasks such as semantic editing and element editing depend on more complex pipelines, and obtaining data for multi-image tasks is even more challenging.
> >
> > ii）Use of High-Quality Data: During the model training phase, we divided the process into Instruction Alignment and Aesthetic Improvement. Higher-quality data helps us achieve a better-quality model.
> >
> > iii）Data Scaling Law: It has been proven that data scaling laws are often simple yet effective, and we are continuously working on data construction.
> >
> > ---
> >
> > **Q6: Model Update**
> >
> > Regarding how the model can be quickly updated, there are two considerations:
> >
> > i）Data-Driven: When the constructed dataset reaches a sufficient scale and high quality, combining and training it with the current state-of-the-art generative models can yield a high-quality model.
> >
> > ii）Model-Driven: Once a foundational editing model is adequately trained, the model itself possesses a certain level of generalization capability. Adapting to new tasks can also be achieved through the rapid application of fine-tuning strategies (such as LoRA, etc.), allowing for plug-and-play support.
> >
> > ---

---

> > > ### Comment · Reviewer_vGPZ · 2024-11-27
> > >
> > > I appreciate the authors for the response. It has addressed most of my concerns and I will maintain my original scores.

---

> > > > ### Author Response · Authors · 2024-11-27
> > > >
> > > > Thanks again for your time and effort in reviewing our work.

---

### Official Review · Reviewer_NXSb · 2024-11-04

**Soundness:** 3
**Presentation:** 2
**Contribution:** 3
**Rating:** 6
**Confidence:** 4

**Summary:**

The paper presents a method to train a unified model for 8 different tasks: Text-guided Generation, Low-level Visual Analysis, Controllable Generation, Semantic Editing, Element Editing, Repainting, Layer Editing and Reference Generation. The idea is intuitive. The main contribution of the paper is the framework for generating paired training data. The source of the data generation comes from two aspects: 1. synthetic generation and 2. from publicly available datasets (LAION-5B). To verify the results of this task, authors also create a new benchmark called ACE Benchmark.

**Strengths:**

1. The dataset generated in this paper is beneficial to the community. This will help other researchers follow this series of research works.
2. A unified model for all tasks is also more efficient compared to have several individual models specific to certain type of tasks.

**Weaknesses:**

1. It seems the author does not have clear discussions on how those tasks affect each other. Are they beneficial to each other? Or some of the tasks are reducing the performance of other tasks? How to select the most reasonable tasks that should be unified with the single model? I believe adding this type of discussion with corresponding experiments will make the paper more solid.

**Questions:**

Indeed, as I mentioned in the weakness, how those tasks affect each other?

---

> ### Author Response · Authors · 2024-11-21
>
> Dear Reviewer NXSb,
>
> Thank you for acknowledging our framework. We address your concerns as follows:
>
> ---
>
> **Q1: Task Interdependence**
>
> Thank you for your suggestions. We have added a discussion on task interdependence in the *"Supplementary Materials, Section IMPLEMENTATION DETAILS, SubSection Task Interdependence"*. The main details are as follows:
>
> In an all-in-one visual generation model, there exist multiple interactions between tasks, similar to that in large language models, and this relationship can be viewed as a complex balancing action.
>
> i) Complementarity between tasks: The combined influence of various tasks can lead to a certain degree of generalized behavior across tasks. For instance, in the style transfer task, our prepared data and training process focus on pixel-aligned global image transfer. However, by incorporating learnings from other tasks related to mask guidance or subject guidance, the model can acquire the ability to perform style transfer in localized areas. (as in Fig. 29)
>
> ii) Competition between tasks: As the scale of tasks increases, the potential for competition also grows, particularly in scenarios where user instructions are ambiguous. For example, when adding the text "APPLE" to an image, it is essential to specify that it is text to be added; otherwise, due to semantic ambiguity, the result may instead involve the addition of an object depicting an apple. (as in Fig. 29)
>
> To achieve optimal performance balance, we first focus on adjusting the data sampling rates for each task in a phased manner during the training process, monitoring this through a validation set. Additionally, more detailed descriptions of instructions are needed in the preparation of training data to prevent semantic confusion between tasks. Through these methods, we aim to ensure that the model can fully leverage the complementarity between different tasks while controlling for any potential negative impacts.
>
> However, the relationships between different tasks still require further exploration to better optimize the model's performance. Future work will also focus on how to effectively evaluate and adjust these influencing factors to achieve a more balanced and comprehensive execution of tasks.
>
> ---
>
> **Q2: Task Selection**
>
> We aim for ACE to encompass all visual generation tasks as possible, which is why we have not conducted a unified consideration specifically focused on reasonable tasks. Specifically, we propose the LCUs to unify various modal conditions, enabling the model to be compatible with different tasks. Additionally, if new visual generation tasks need to be supported, they can also be processed and fine-tuned accordingly through this paradigm.
>
> ---

---

> > ### Comment · Reviewer_NXSb · 2024-11-27
> >
> > I thank the authors for the response. I believe an in-depth analysis of the relation between tasks would strengthen the paper, so Nevertheless, the paper has some contributions to the community. Thus, I will keep my original score.

---

> > > ### Author Response · Authors · 2024-11-27
> > >
> > > Thank you for acknowledging our contributions, as well as the time and effort invested.

---

### Author Response · Authors · 2024-11-21

Dear all,

We would like to express our gratitude to our reviewers for their valuable comments. For positive comments,
- significant contribution (R-vGPZ, R-7X1X),
- unified & broad task support & efficient framework (R-NXSb, R-vGPZ, R-bd3V),
- systematic and complete work (R-vGPZ, R-vndP),
- comprehensive evaluation benchmark (R-vGPZ, R-bd3V, R-7X1X),
- well writing and drawing with excellent reading experience (R-7X1X)

we appreciate them and will carry them forward.

We would like to further clarify our goal and contributions, address the common concerns raised by the reviewers, and outline the revisions made to the manuscript in response to these comments.

**1. Goal**

We aim to create an all-in-one model that supports a wide range of visual generation and editing tasks, which we have named ACE: an All-round Creator and Editor. Currently, it covers eight basic types: Text-guided Generation, Low-level Visual Analysis, Controllable Generation, Semantic Editing, Element Editing, Repainting, Layer Editing, and Reference Generation. To achieve this, we define a universal LCU input paradigm, design specific data construction pipelines, and propose a comprehensive evaluation benchmark. We hope to continually expand task capabilities and improve generation quality, providing momentum for the development of the open-source community.

**2. Task Interdependence**

We add a discussion about this in the *"Supplementary Materials, Section IMPLEMENTATION DETAILS"*. Below, we briefly describe the issue:
In an all-in-one visual generation model, there exist multiple interactions between tasks, similar to that in large language models, and this relationship can be viewed as a complex balancing action. To achieve an optimal performance balance, we handle the data preparation and model training processes to ensure that the model can fully leverage the complementarity between different tasks while controlling for any potential negative impacts.

**3. Design choices**

We add an additional section in the *"Supplementary Material, Section ARCHITECTURE DESIGN"* to clarify our design considerations and provide relevant visual analyses, which, in brief, include the following modules:
Long-context Attention Block is specifically designed to handle input image sequences of varying lengths, addressing the limitations of the conventional attention block employed in DiT, which cannot effectively manage sequences of disparate lengths.
Image Indicator Embeddings facilitate the alignment between images in the sequence and their corresponding mentions within the text prompt.

**4. More implementation details**

Based on the reviewers' comments, we have supplemented relevant content in the *"Supplementary Material, Section IMPLEMENTATION DETAILS"*, including a further description of the training process and adding the corresponding parts of data preprocessing, computational efficiency, checkpoints evaluation, and visualization of the editing process to address the reviewers' concerns.

For other concerns, we address them in the respective comments to the reviewers.


Thanks and best regards,

Authors of Submission 8711

---

### Comment · Area_Chair_kfBz · 2024-11-28
**Reviewer feedback and discussion**

Dear Reviewers,

As the discussion period will end next week, please take some time to read the authors' rebuttal and provide feedback as soon as possible. For reviewers vndP, bd3V, and 7X1X, did the author address your concerns, and do you have further questions?

Thanks,

Area Chair

---

### Meta-Review · Area_Chair_kfBz · 2024-12-19

**Metareview:**

This paper proposes an all-in-one model that supports a wide range of visual generation and editing tasks. Reviewers recognize the contribution of the unified framework and extensive evaluations. Questions are raised regarding design choices, more analysis, and experiments. The authors addressed most of the concerns during the rebuttal, and all reviewers give positive scores. Therefore, the area chair recommends accepting the paper.

**Additional Comments On Reviewer Discussion:**

The authors addressed the reviewer questions on design choices, more analysis (such as task independence), and experiments during the rebuttal. Two reviewers replied in the rebuttal that there concerns are addressed. The other three reviewers did not reply (although the authors and area chair have asked these reviewers for multiple times). The area chair checked the reviewer's response and believes that the authors have adequately addressed major concerns proposed by those reviewers.

---

### Decision · Program_Chairs · 2025-01-22

Accept (Poster)